

# Nocturnal low-level clouds in the atmospheric boundary layer over southern West Africa: an observation-based analysis of conditions and processes

Bianca Adler[1], Karmen Babić[1], Norbert Kalthoff[1], Fabienne Lohou[2], Marie Lothon[2], Cheikh Dione[2], Xabier Pedruzo-Bagazgoitia[3], and Hendrik Andersen[1]

[1]Institute of Meteorology and Climate Research, Karlsruhe Institute of Technology (KIT), Karlsruhe, Germany
[2]Laboratoire d'Aérologie, Université de Toulouse, CNRS, UPS, Toulouse France
[3]Meteorology and Air Quality Group, Wageningen University and Research, Wageningen, the Netherlands

**Correspondence:** Bianca Adler (bianca.adler@kit.edu)

**Abstract.** During the West African summer Monsoon season, extended nocturnal stratiform low-level clouds (LLC) frequently form in the atmospheric boundary layer over southern West Africa and persist long into the following day affecting the regional climate. A unique data set was gathered within the framework of the Dynamics-Aerosol-Chemistry-Cloud-Interactions in West Africa (DACCIWA) project, which allows, for the first time, for an observational analysis of the processes and parameters decisive for LLC formation. In this study, in situ and remote sensing measurements from radiosondes, ceilometer, cloud radar and energy balance stations from a measurement site near Savè in Benin are analyzed amongst others for 11 nights. The aim is to study LLC characteristics, the intra-night variability of boundary layer conditions and physical processes relevant for LLC formation, as well as to assess the importance of these processes. Typical nocturnal phases are identified and mean profiles are calculated for the individual phases revealing pronounced differences: a stable surface inversion, which forms after sunset, is eroded by differential horizontal cold air advection with the Gulf of Guinea maritime inflow, a cool air mass propagating northwards from the coast in the late afternoon and the evening, and shear-generated turbulence related to a nocturnal low-level jet. The analysis of the contributions to the relative humidity changes before the LLC formation reveals that cooling in the atmospheric boundary layer is decisive to reach saturation, while moisture changes play a minor role. We quantify the heat budget terms and find that about 50 % of the cooling prior to the LLC formation is caused by horizontal cold air advection, roughly 20 % by radiative flux divergence and about 22 % by sensible heat flux divergence in the presence of a low-level jet. The outcomes of this study contribute to the development of a conceptual model on LLC formation, maintenance and dissolution over southern West Africa.

*Copyright statement.* TEXT



# 1 Introduction

Nocturnal stratiform low-level clouds (LLC) frequently form in the atmospheric boundary layer (ABL) over southern West Africa during the West African summer Monsoon season. These LLC have typical cloud-base heights (CBH) of only a few hundred meters above ground (Schrage and Fink, 2012; Kalthoff et al., 2018) and cover an area of about 800 000 km$^{-2}$ (van der Linden et al., 2015). The LLC form during the night and persist long into the following day (Kalthoff et al., 2018). They thus affect the energy balance at the Earth's surface, the diurnal cycle of the ABL and the regional climate (Knippertz et al., 2011; Hannak et al., 2017).

Most of our knowledge on the processes relevant for the formation, maintenance and dissolution of the LLC is based on numerical simulations, as high-quality observational data are scarce in this region. Studies by Schrage and Fink (2012), Schuster et al. (2013), Adler et al. (2017) and Deetz et al. (2018) suggest that processes spanning from the microscale to the synoptic scale are important such as horizontal cold air advection, orographic lifting, lifting related to gravity waves and shear-generated vertical mixing underneath the axis of a nocturnal low-level jet (LLJ). There is evidence that the horizontal cold air advection is related to the south-westerly Monsoon flow which transports maritime air from the Gulf of Guinea northwards over land. The daytime conditions at the coast are characterized by a superposition of the Monsoon flow and a sea breeze. During the Monsoon season, the strong south-westerly Monsoon flow dominates and makes it difficult to distinguish between both (Bajamgnigni Gbambie and Steyn, 2013). During the day, the south-westerly flow is decelerated over land leading to convergence along a line parallel to the coast often associated with moist convection (Knippertz et al., 2017; Parker et al., 2017) and rainfall (Maranan et al., 2018). Model studies indicate that the convergence zone which separates the cool maritime air in the south from the warmer air in the convective ABL over land is rather stationary and located at several tens of kilometers distance from the coast until the late afternoon (Adler et al., 2017; Deetz et al., 2018). In the late afternoon and early evening, the cool maritime air starts to propagate inland and reaches distances of more than 100 km. Grams et al. (2010) investigate a sea breeze front which is stationary at the coast of Mauritania during daytime and propagates inland in the evening. These authors relate the stationarity to a balance between horizontal advection of cool maritime air and turbulence in the convective ABL over land and the inland propagation to the decay of turbulence in the late afternoon. As the maritime air which propagates northwards in our investigation area in southern West Africa has its origin over the Gulf of Guinea, we call this feature *Gulf of Guinea maritime inflow* and refer to it as *maritime inflow* hereafter for the sake of brevity.

To evaluate the hypotheses from previous studies and to enhance our understanding of the physical processes relevant for LLC, high-quality comprehensive observations in the coastal region of southern West Africa were urgently needed. Therefore, a concerted measurement campaign was conducted in summer 2016 within the framework of the Dynamics-Aerosol-Chemistry-Cloud Interactions in West Africa (DACCIWA) project (Knippertz et al., 2015). The meteorological measurements were airborne (Flamant et al., 2018) as well as ground-based at three supersites in Ghana, Benin and Nigeria (Kalthoff et al., 2018) and depict the most comprehensive data set for this region so far. An overview of the meteorological large-scale conditions during the campaign is given by Knippertz et al. (2017).



Based on the observational data gathered at Savè, the supersite with the most comprehensive instrumentation, a series of analysis have been conducted on the LLC: Babić et al. (2018) present - for the first time - a detailed analysis of the diurnal cycle of LLC for a case study of a typical night (7-8 July, intensive observation period (IOP) 8) and identify physical processes and factors which control the formation, maintenance and dissolution of LLC at Savè. Based on the dynamical and

thermodynamical conditions in the ABL during IOP 8, these authors identify different phases, which are outlined in Fig. 1: the stable phase describes a period after sunset when the horizontal wind is weak and a surface inversion forms. With the arrival of the maritime inflow a few hours after sunset, dynamic and thermodynamic conditions in the ABL change and are then characterized by a LLJ wind profile, lower temperature and lower static stability (jet phase). Eventually, LLC form (stratus phase). About 1 h after sunrise, the cloud base starts rising due to the evolution of the convective ABL (convective phase). Babić

et al. (2018) determine the contributions of temperature and humidity changes to the relative humidity changes and quantify the heat budget terms for the different phases during IOP 8 using radiosoundings. Dione et al. (2018) perform a detailed statistical analysis on the characteristics of the LLC and the low-level atmospheric dynamics using mainly data from continuously running remote sensing instruments for a 41-days period. While the studies of Babić et al. (2018) and Dione et al. (2018) either look at the diurnal cycle during one case study or at mean quantities during a longer measurement period, the present analysis focuses

on 11 IOP nights. As radiosoundings were performed in short temporal intervals of 1 to 1.5 h throughout the IOP nights, we are able to perform an analysis for these nights in a manner consistent with the methods used by Babić et al. (2018). This allows us to generalize some of the findings from the single IOP case study. Besides the generalization of process relevance for the formation of LLC, we aim to characterize the LLC as well as to investigate the intra-night variability of ABL conditions. The research questions to be answered are: (i) What are the temporal and spatial characteristics of LLC? (ii) How do ABL

conditions change during the different nocturnal phases? (iii) What dominates the relative humidity changes and heat budget? (iv) How do the processes involved in (iii) vary with height and from night to night?

In Sect. 2, the observational data used and the methods applied to derive different LLC characteristics as well as to estimate the heat budget terms are described. Section 3 includes LLC characteristics and Sect. 4 the conditions in the nocturnal ABL for the different phases. In Sect. 5, relative humidity changes and heat budgets terms are investigated and processes resulting

in LLC formation are assessed. In Sect. 6, the observed processes are discussed in comparison to recent studies, followed by a summary and conclusions in Sect. 7.

## 2  Data used and methods

### 2.1  Measurement site and instrumentation

The supersite at Savè (166 meters above mean sea level (m m.s.l.)) was located in Benin about 185 km inland from the

coast (Fig. 2b). The terrain in the immediate surrounding of the site is rather flat, while higher terrain up to 500 m m.s.l. is found north and east of it. During the 7 weeks of the ground-based measurement campaign comprehensive in situ and remote sensing measurements were conducted. A detailed overview of the atmospheric conditions during the whole campaign period is given by Kalthoff et al. (2018) and the complete instrumentation including information on the manufacturers is described by



Bessardon et al. (2018). In this study, we use data from the continuously running ceilometer (measuring backscatter profiles from which information on CBH is derived) and cloud radar (measuring radar reflectivity profiles from which information on cloud-top heights (CTH) is derived) to obtain information on the LLC characteristics at Savè. Spatio-temporal information on LLC in a larger area around Savè (0-4° E and 5.5-10° N, dashed box in Fig. 2b) is obtained from the Spinning Enhanced

Visible and Infrared Imager (SEVIRI) sensor (Schmetz et al., 2002). Two energy balance stations provide the near-surface radiation and energy balances as well as meteorological parameters over two types of land use (fallow and corn) and an ultra-high-frequency (UHF) wind profiler and a Doppler lidar are used to obtain information on the horizontal and vertical wind. Ceilometer, cloud radar, one of the energy balance stations and the Doppler lidar are part of the mobile integrated atmospheric observation system "KITcube" deployed by the Karlsruhe Institute of Technology (Kalthoff et al., 2013).

During the 7 weeks of the campaign, 15 IOPs in total were conducted (Table 1). Every IOP lasted from the late afternoon throughout the night until the afternoon of the following day to capture the whole diurnal cycle of the LLC. Although we aimed to perform IOPs during synoptically undisturbed nights without any mesoscale convective systems, the conditions during 3 of the 15 IOPs (IOPs 2, 12 and 13) were disturbed by rain at or near the supersite, preventing the evolution of "typical" LLC. This is why we exclude these IOPs from this analysis. As no LLC existed during IOP 10 at Savè, this leaves 11 IOPs for the

analysis of the relation between ABL conditions and LLC (the used IOPs are highlighted in Table 1). During IOPs, radiosondes were released at Savè one hour before the nominal times in 6 hourly intervals, i.e. the first radiosounding was released at 1700 UTC (the local standard time in Benin is UTC+1), in order to be synchronous with the soundings at operational radiosonde stations. In between, so-called frequent radiosondes were launched reaching maximum heights of around 1500 meters above ground level (m a.g.l.) to get a higher temporal resolution of the ABL conditions. These sondes are attached to two ballons

of different volume, whereas the line to the larger balloon is cut after a preset time of ascent (corresponding to around 1500 m a.g.l.) initiating a controlled descent of the sonde (Legain et al., 2013). This method allows a re-use of the sonde and short sounding intervals. During IOP 1-6, frequent radiosondes were launched in hourly intervals starting at 2100 UTC. After IOP 7, the sounding schedule was changed to 1.5-h intervals starting at 1830 UTC to better resolve the early evolution of the nocturnal ABL. An impression of the radiosonde schedule during individual IOPs can be obtained from Fig. 3 with each coloured column

indicating one sounding centered on the launch time.

In addition and as part of the DACCIWA radiosonde campaign (Flamant et al., 2018) radiosoundings in up to 6-h intervals were launched at three stations along the coast, at Abidjan (Ivory Coast), Accra (Ghana) and Cotonou (Benin) (locations are given in Fig. 2b).

## 2.2  LLC characteristics

### 2.2.1  Cloud-base and cloud-top height detection

From the ceilometer backscatter profiles with 1-min resolution, up to three CBH are obtained using the manufacturer algorithm which is based on a threshold method (we only use the lowest CBH in this study). The cloud radar deployed at Savè is dual-polarized, i.e. it is possible to distinguish between hydrometeors and other targets. The classification uses the linear



depolarization ratio as well as multi-peak moments and is described in detail in Bauer-Pfundstein and Görsdorf (2007). From the radar reflectivity of hydrometeors we estimate the CTH for 5-min averaged reflectivity profiles applying a threshold of -35 dBz, i.e. reflectivities larger than -35 dBz are considered as clouds. Examples of ceilometer backscatter profiles and cloud radar reflectivity profiles with estimated CBH and CTH are shown in Fig. 3 in Babić et al. (2018). Beside information on dynamic and

thermodynamic conditions, estimates of the vertical extent of clouds can be obtained from relative humidity profiles measured with radiosondes. The algorithm uses a threshold of 99 % and is described in Kalthoff et al. (2018). The CTH used in this study are a combination of estimates from cloud radar and radiosonde measurements. During some soundings, discrepancies occur between CTH estimated from cloud radar and radiosonde measurements which are likely related to condensation of water vapour on the sensors when the radiosonde flies through a cloud (more details on this are given in Babić et al. (2018)).

### 2.2.2 Estimation of LLC onset time

Before estimating onset times of LLC from ceilometer a clear definition of LLC must be stated. Do we require a constant CBH and a complete coverage of the sky or do we allow some variability in CBH and coverage? From the overview of CBH in Kalthoff et al. (2018), we expect LLC bases mainly to occur in the lower 600 m and consequently choose this layer to estimate cloud-base fraction from ceilometer. As CBH are available every minute from the ceilometer we calculate the cloud-base

fraction every minute for the subsequent 60 min. The onset time is then defined as the first point in time when a cloud-base is detected in the lower 600 m and the subsequent cloud-base fraction is higher than a respective threshold. During all IOP nights cloud-base fractions reach 100 %. During some nights, cloud-base fraction shifts almost instantly from 0 % to 100 %, while during other nights, lower cloud-base fractions precede the complete coverage by several hours. To take this into account, we introduce the stratus fractus phase before the stratus phase, in addition to the phases identified by Babić et al. (2018) for

IOP 8 (Fig. 1). We choose two thresholds, 50 and 95 %, to detect the onset of stratus fractus and stratus, respectively. For the estimation of tendencies and contributions we look at the period before the onset of stratus fractus, to avoid the impact of phase changes, while we investigate the modification of the ABL conditions by the LLC for the stratus phase only, as we expect a clearer signal from this phase.

### 2.2.3 Detection of the horizontal distribution of LLC

Detecting LLC during the night from the geostationary SEVIRI sensor is challenging, as the temperature at the cloud top is very close to the surface temperature in cloud-free regions, making them nearly indistinguishable from the surface in the infrared channels. In the absence of high or mid-level clouds, which are obscuring the LLC from the view of the satellite-borne sensor, the brightness temperature difference of the thermal-infrared channel at 10.8 µm and the middle-infrared channel at 3.9 µm are used to illustrate the LLC during the night. Higher-level clouds are masked out by applying a brightness temperature

threshold of 283 K to the 10.8 µm channel, i.e. cloud tops above around 2500 m m.s.l. are masked. More details on the method are given in Babić et al. (2018).



## 2.3 Estimation of heat budget terms

The tendency of the mean potential temperature (TOT) is generally influenced by various processes such as horizontal (HADV) and vertical (VADV) advection, radiative flux divergence (RAD), sensible heat flux divergence (TURB) and phase changes (SQ) (e.g. Stull, 1988):

$$\underbrace{\frac{\partial \Theta}{\partial t}}_{\text{TOT}} = \underbrace{-u\frac{\partial \Theta}{\partial x} - v\frac{\partial \Theta}{\partial y}}_{\text{HADV}} \underbrace{-w\frac{\partial \Theta}{\partial z}}_{\text{VADV}} + \underbrace{\frac{1}{\rho c_p}\frac{\partial Q}{\partial z}}_{\text{RAD}} \underbrace{-\frac{1}{\rho c_p}\frac{\partial H}{\partial z}}_{\text{TURB}} \underbrace{-\frac{LE}{\rho c_p}}_{\text{SQ}} \tag{1}$$

with the mean potential temperature, $\Theta$, the mean wind components, $u$, $v$ and $w$, the mean air density, $\rho$, the specific heat capacity at constant pressure, $c_p$, the sensible heat flux, $H$, the latent heat of vaporization of water, $L$, and the evaporation rate $E$. The estimation of the different heat budget terms from observations is challenging and requires some assumptions. The easiest term to derive is TOT, which we calculate directly from radiosonde profiles. As we only consider periods without LLC, we can neglect SQ in the budget. Sun et al. (2003) calculate radiative and sensible heat flux divergence from tower measurements for the nocturnal ABL and find that even small vertical or horizontal temperature difference can contribute significantly to TOT. The measurement accuracy of mean vertical velocity required to estimate VADV is hardly achieved by in situ wind measurements and even less by remote sensing instruments. This is why we cannot estimate VADV in this study. This leaves us with HADV, RAD and TURB, for which we now describe the methods we used for their estimations. The estimations are done for different periods (Fig. 1). The period for HADV comprises the stable and jet phases, while TURB is calculated during the jet phase only and RAD and TOT are estimated for both time periods.

### 2.3.1 Horizontal temperature advection

The large-scale dynamic and thermodynamic conditions in the investigation area in southern West Africa are characterized by the south-westerly Monsoon flow and a meridional temperature gradient with lower temperatures over the Gulf of Guinea. As outlined in the introduction, there is evidence that cool maritime air is transported further inland with the Monsoon flow during the late afternoon and night – a feature which we call maritime inflow. During the IOP days studied here, we detect an concurrent increase of wind speed with the profile showing a LLJ structure and a decrease of temperature in a layer of several hundred metres depth during the first half of the night, which we interpret as the arrival of the maritime inflow at Savè. The changes in atmospheric conditions during this arrival are nicely illustrated for IOP 8 in Babić et al. (2018). In the present study we use radiosoundings to detect the changes in temperature and wind profiles and manually distinguish between the different phases as well as allocate the radiosonde profiles to the phases. Note that Dione et al. (2018) find some differences in the onset times of the LLJ and the maritime inflow when analysing the whole campaign period. These differences may depend on the higher temporal resolution of the data from continuous remote sensing instruments as well as on the criteria applied by these authors to detect the onset times.

In order to estimate horizontal cold air advection related to the maritime inflow from the available observations, several assumptions are necessary, which are illustrated in the schematic diagram in Fig. 4. To estimate the meridional temperature difference, we use radiosoundings performed at the three coastal stations and at Savè in the late afternoon (station locations in



Fig. 2b). We assume that the temperature distribution is homogeneous along the coast and that the zonal temperature gradient and wind component are small and make the following estimates for a meridional cross section through Savè. The difference of the mean temperature in the late afternoon between the coast and Savè is more than 3 K on the average (Table 1). Information on the meridional temperature distribution between Savè and the coast are obtained from aircraft flights, when horizontal legs

were flown in the ABL at some angle to the coast line in the afternoon (not shown). These measurements indicate that the aircraft passes through the maritime inflow as well as through the convective ABL over the land during these flights. In the convective ABL, the temperature is rather horizontally homogeneous, while it decreases gradually towards the coast within a certain distance from the coast. We interpret this distance as the maximum inland propagation of the maritime inflow at that time (right edge of yellow box in Fig. 4). This means that the temperature decreases gradually within the maritime inflow and

we assume a linear increase of temperature in south-north direction in the maritime inflow and a constant temperature in the convective ABL (red curve in Fig. 4). Motivated by model studies by Adler et al. (2017) and Deetz et al. (2018), we expect the maximum inland propagation to be somewhere between 50 and 125 km inland from the coast in the late afternoon.

For the cooler air of the maritime inflow to be able to produce a cooling at Savè, the maritime inflow has to propagate far enough inland to reach the site. To estimate the maximum distance from which the air mass measured at Savè before the LLC

onset may originate (left edge of the dashed area in Fig. 4), we estimate the propagation speed from radiosoundings. Therefore, we average the meridional component of the coastal wind profile in the afternoon and of the wind profile at Savè after the maritime inflow arrives, $v$, and assume that the maritime inflow propagates with the maximum southerly wind component of the averaged vertical profile. If the propagation speed is high enough and the LLC onset late enough, the air in the maritime inflow is able to reach Savè and to contribute to the cooling before the LLC onset. For example, assuming a propagation speed

of 5 m s$^{-1}$, a LLC onset at Savè at midnight and a start time of the further inland propagation of the maritime inflow at 1600 UTC, the air mass passing Savè can originate at a maximum distance of 41 km inland from the coast. With an assumed maximum inland propagation of the maritime inflow of 75 km during the day, parts of the air mass passing Savè before the LLC onset are of maritime origin and have a by maximum $\Delta\Theta$ lower temperature than the air in the convective ABL. Based on the assumptions for the maximum distance and the linear temperature change in the maritime inflow (Fig. 4), HADV is

calculated as $-v\Delta\Theta\,\Delta y^{-1}$ using data from all three coastal stations (Abidjan, Accra and Cotonou), if available, as well as four different maximum propagation distances (50, 75, 100 and 125 km).

### 2.3.2 Radiative flux divergence

To estimate the contribution of RAD to the temperature tendency at Savè, we apply the Santa Barbara DISORT Atmospheric Radiative Transfer (SBDART) model (Ricchiazzi et al., 1998) to the mean profile averaged for the time period between 1700

UTC and the formation of clouds as well as for the jet phase (Fig. 1) using a mean aerosol optical depth measured with the sun photometer at Savè and averaged for the months June and July 2016. More details on the radiative transfer model and on the used input parameters can be found in Babić et al. (2018).



### 2.3.3 Turbulent heat flux divergence

During the night, the absolute value of the sensible heat flux is usually at its maximum at the surface and decreases with height (e.g. Sun et al., 2003). To estimate TURB from the existing measurements we need to make an assumption for the height where the sensible heat flux vanishes. In the presence of a surface inversion, this height is usually assumed to be the top of the surface

inversion. If a LLJ is present, we assume that the sensible heat flux vanishes at the height of the LLJ axis (as vertical wind shear vanishes). Both cases are present at Savè i.e. the stable phase and jet phase. Due to the insufficient number of radiosonde profiles during the stable phase (in particular during the IOPs 1-6), we estimate TURB for the jet phase only (Fig. 1). For this estimation we use measurements of the surface sensible heat flux at both energy balance stations and horizontal wind profiles to detect the height of the LLJ axis. The surface sensible heat flux is averaged over the considered time period and the bulk

cooling by TURB is then estimated for the layer below the LLJ axis.

### 3 LLC characteristics

This section overviews the vertical distribution and temporal evolution of LLC at Savè using ground-based remote sensing instruments as well as the horizontal distribution of LLC using satellite images.

The combination of ceilometer and cloud-radar measurements allows to obtain information on the vertical extent of the

LLC. In Fig. 3, cloud base from ceilometer and cloud top from cloud-radar measurments are shown as red dots and circles, respectively, and Fig. 5a contains statistical information on CBH and CTH during the stratus phase for the individual IOPs. The median CBH ranges from 70 m to 450 m a.g.l. and the median CTH from 370 to 870 m a.g.l. When averaging the median heights and the vertical extents for all IOPs, CBH = 250±120 m a.g.l., CTH = 590± 170 m a.g.l. and a vertical extent of 340±80 m result. This confirms the vertical extent of simulated LLC (Schuster et al., 2013; Adler et al., 2017) and agrees with

the median CBH and CTH pointed out by Dione et al. (2018) using all days of the DACCIWA campaign. During IOP 1, 5 and 6, the median CBH is below 130 m a.g.l., while it is above 200 m a.g.l. for the other IOPs. This is in agreement with the two layers found favourable for CBH occurrence at Savè by Kalthoff et al. (2018) (see their Fig. 5) and will be taken up again in Sect. 5.3.

From the CBH measured by the ceilometer the onset time of LLC is derived applying two thresholds for the cloud-base

fraction (Sect. 2.2.2). The onset of stratus fractus and stratus is indicated by the unfilled green and black markers, respectively, in Figs. 3 and 5b. In Fig. 3 these markers are placed at the top of each plot. During some nights long periods with stratus fractus precede stratus (e.g. IOPs 4 and 14), while no stratus fractus at all is detected during other nights (e.g. IOPs 1, 5, 6 and 8). Both onset times vary considerably for the individual IOPs. Stratus fractus occurs as early as 2100 UTC (IOPs 4 and 14) and stratus onset times range from 2200 UTC on IOP 9 to as late as 0445 UTC on IOP 15. In the following we define the LLC onset time

as the onset time of stratus fractus or of stratus, if no stratus fractus is present. Note that Dione et al. (2018) detect the onset time of LLC from an infrared camera which took an image of some fraction of the sky every 2 min. As this method strongly depends on the homogeneity of CBH, cloud-base cover, cloud density and vertical extent, derived times sometimes differ from





the stratus onset detected by ceilometer. The root-mean-square error between both methods is approximately 80 min for IOP days.

The horizontal distribution and evolution of LLC is investigated using satellite images (Fig. 6) created with the method described in Sect. 2.2.3. Unfortunately, only during 4 IOPs the LLC in the surroundings of Savè are not masked by high- or mid-level clouds, i.e. IOPs 3, 4, 7 and 8. Despite the small sample size, the spatial LLC characteristics show a large variability with respect to the location of first LLC formation and to the directions of the subsequent growth:

   i. During IOP 3 LLC form already at 2200 UTC over the higher terrain north and north-east of Savè (Fig. 2b) and remain more or less stationary until around 0330 UTC expanding only little (Fig. 6a). After that, the LLC suddenly start to expand to the south-west until they cover most of the domain including Savè. This is in agreement with the onset of LLC at around 0400 UTC at Savè (Figs. 3b and 5b).

   ii. In the evening of IOP 4, some mid- and high-level clouds obscure the lower levels in many parts of the domain, but it is nevertheless evident that LLC exist already at 2100 UTC east of Savè over the higher terrain (Fig. 6b). In the following hours the mid- and high-level clouds move westwards allowing to detect the growth of the LLC towards the west affecting Savè. These LLC have quite scattered cloud bases (Fig. 3c). Between around 0030 and 0230 UTC a spatial gap occurs in the LLC deck right above Savè (Figs. 3c and 6b). After that the LLC cover most of the domain for the rest of the night (Fig. 6b) and the CBH are then rather homogeneous as visible at Savè (Fig. 3c).

   iii. During IOP 7, the domain is mostly cloud-free until 2300 UTC (Fig. 6c). Then, LLC start forming at several locations in the domain and grow in the subsequent hours, also occuring at Savè after midnight (Fig. 3f). After around 0100 UTC, high-level clouds in a layer between around 11 and 13 km a.g.l. (as visible in cloud-radar measurements at Savè) gradually move in from the east and cover the domain preventing the analysis of the further evolution of the LLC.

   iv. During IOP 8, first LLC form south-west and east of Savè after around 2200 UTC (Fig. 6d). In the following hours both patches of LLC expand, the one in the south-west reaching Savè at midnight (Fig. 3g). After 0030 UTC both patches grow together forming a roughly zonal band. Subsequently, this band grows in all directions and LLC cover most of the domain at sunrise.

Overall, we can identify two types of horizontal LLC expansion: during IOPs 3, 4 and 8, the LLC grow to the upstream side, i.e. towards the direction of the mean south-westerly flow, as well as to the downstream side, i.e. away from the direction of the mean flow. During IOP 7, LLC form and grow at many locations at the same time with no clear direction being distinguishable. This suggests that different mechanisms are involved, some of those are assessed in Sect. 5.3.

## 4  Intra-night variability of ABL conditions

This section characterizes the mean nocturnal ABL conditions during different phases of the night - these are the LLC-free stable and jet phases as well as the stratus phase (Fig. 1). The profiles which we use for the averaging during the individual



phases are indicated at the bottom of each plot in Fig. 3 by the crosses. To take into account the large variability of CBH during the individual IOPs (Fig. 5a), we normalize the profiles with the median CBH of each IOP. Despite the large day-to-day variability in the wind speed profiles (Fig. 3), there is a clear signal in the mean normalized profiles (Fig. 7a): during the stable phase, the mean wind speed is lowest with around 3 m s$^{-1}$ and little variation with height, while the LLJ shape is clearly visible during the jet and stratus phases. During the jet phase, the LLJ axis with a maximum value of more than 7 m s$^{-1}$ is near the height where cloud bases exist later on. In the presence of LLC, the LLJ axis shifts upwards to around 1.6 z/CBH and the maximum decreases by about 1 m s$^{-1}$. The mean potential temperature decreases in the course of the night leading to an up to 4 K cooler atmosphere during the jet phase than during the stable phase and a decrease of about 1 K from the jet to the stratus phase (Fig. 7b). Specific humidity changes little during the stable phase and jet phase, while it is about 1 g kg$^{-1}$ lower during the stratus phase (Fig. 7c). The strongest changes occur up to around 2 z/CBH. The stable phase is characterized by a shallow surface inversion and a weakly stably stratified residual layer above (Figs. 7b and d). During the jet phase the surface inversion is eroded (Sect. 5.2.2) and static stability becomes more constant with height but has quite a large standard deviation. During the stratus phase, static stability below and also above the CBH decreases compared to the jet phase. This is likely driven by longwave cooling at cloud top, which leads to mixing. The top of the layer with reduced stability coincides with the LLJ axis during the stratus phase. This suggests that the upward shift of the inversion due to the reduced stability in the presence of clouds causes the LLJ axis to shift upwards, which agrees with the results of Babić et al. (2018) and Dione et al. (2018).

## 5 Changes in atmospheric conditions and processes leading to LLC formation

### 5.1 Relative humidity changes

For LLC to form, the ABL has to be saturated, i.e. relative humidity has to reach 100 %. In order to understand why saturation is reached, i.e. due to cooling or moistening, we calculate the contribution of temperature and humidity changes to the relative humidity changes between the late afternoon and just before the onset of LLC (the formula is given in Babić et al. (2018) and for the placement of the period see Fig. 1). For this we use the radiosonde profiles at 1700 UTC and the last profile before the LLC onset at Savè, i.e. the elapsed time varies between 4 h for IOPs 4, 9, 11 and 14 and 10 h during IOPs 1, 3 and 15. The mean profiles of relative humidity changes and the individual contributions for this period are shown in Fig. 8a. On the average, relative humidity increased by about 25 % near the surface decreasing linearly with height up to around 750 m a.g.l. The shape of this profile is mainly due to the linear increase of relative humidity with height in the ABL in the late afternoon. Nearly 100 % of the relative humidity increase are related to cooling, while moistening contributes only little (Fig. 8a).

To investigate this further, we split the period in two parts indicated in Fig. 1: one period between the late afternoon and just after the arrival of the maritime inflow (P1) and one period after the arrival and before the LLC onset (P2). During P1, 5 % of the relative humidity increase in lower layer are caused by moistening, which is a contribution of about 25 % (Fig. 8b), while a drying of about 5 % in lower layers occurs during P2 (Fig. 8c). This means that moistening contributes to the relative humidity increase before and at the arrival of the maritime inflow at Savè. Once Savè is within the maritime inflow air mass, specific humidity decreases working against the cooling with respect to the relative humidity change. Independent of the considered




period, cooling is decisive to achieve saturation in the nocturnal ABL. This is in qualitative agreement with the results for IOP 8 in Babić et al. (2018). The small moisture changes indicate that the moisture content in the maritime air mass is roughly the same as in the continental ABL, i.e. no pronounced zonal moisture gradient prevails between Savè and the coast. This is likely related to the relatively low sea surface temperature of the Gulf of Guinea and high evapotranspiration from the dense

vegetation over land.

## 5.2 Heat budget terms

From the analysis of relative humidity changes in the previous section, we know that cooling is the key factor for LLC formation. Consequently, we continue with an analysis of the different terms of the heat budget at Savè (Eq. 1). We start with the estimation of TOT, RAD and HADV for the period from the late afternoon until the onset of LLC, followed by an analysis of

TURB during the jet phase (different periods are filed in Fig. 1) .

### 5.2.1 Heat budget estimates for the period from the late afternoon until LLC onset

We calculate the profiles of TOT between the 1700 UTC radiosonde profile and the last profile before the LLC occur at Savè and average this over all IOPs (black curve in Fig. 9a). The cooling is strongest near the surface and decreases more or less linearly up to around 750 m which is consistent with the profiles of relative humidity changes (Fig. 8a). The variability of

RAD between the individual IOPs is rather small and the mean cooling rates are in the order of -0.15 K h$^{-1}$ near the surface increasing with height to around -0.1 K h$^{-1}$ (green line in Fig. 9a). The mean profile of HADV averaged over all IOPs (red line) resembles the horizontal wind profiles used for the calculations with maximum values between 200 and 400 m a.g.l., i.e. around the level of the LLJ axis. Above around 400 m a.g.l. the sum of RAD and HADV is nearly equal to TOT leaving a small positive residuum only. Below this height the residuum gains high negative values with maximum values near the surface, i.e.

the actual cooling is stronger than the contributions by HADV and RAD.

To investigate the night-to-night variability of TOT, HADV and RAD, we vertically average the different terms up to the level where TOT becomes larger than $\mathrm{TOT_{max}}$ e$^{-1}$ with e being Euler's number and $\mathrm{TOT_{max}}$ the maximum cooling for each IOP (mean and median of this level are 530 and 475 m a.g.l., respectively). The absolute vertically averaged cooling rates are shown in Fig. 9b. TOT varies between around 0.4 and 0.8 K h$^{-1}$, while the variability of RAD is small and absolute values

are around 0.1 K h$^{-1}$ as expected from the profiles. HADV varies considerably and reaches values from around 0.1 to 0.4 K h$^{-1}$. The standard deviation of HADV which results from the usage of the three coastal stations and four different maximum propagation distances reaches up to around 60 % of the mean HADV. On the average, RAD can explain about 20 % and HADV about 50 % of the observed cooling rate. This means that about 30 % of the cooling between the late afternoon and the LLC onset are caused by other processes. From the mean profiles in Fig. 9a it is evident that most of the missing cooling occurs

below around 400 m a.g.l. We expect that a large part of the missing cooling is related to TURB, which is further investigated in the following section.





### 5.2.2 Sensible heat flux divergence during the jet phase

The period before the LLC form can roughly be divided into two phases, i.e. the stable phase and the jet phase (Fig. 1). These phases are illustrated in the following using the example of IOP 15 (Fig. 10): the stable phase lasting from around sunset to 2000 UTC is characterized by the evolution of a shallow surface inversion (Fig 10a), relatively weak winds (Fig. 10a, b), low

TKE values near the surface (Fig. 10c), strong stability as indicated by large Flux and Bulk-Richardson numbers (Fig. 10d) and small negative sensible heat fluxes (Fig. 10e). The Bulk-Richardson number is calculated for the layer between the surface and 200 m a.g.l. as this depicts the height of the mean wind speed maximum (not shown). With the arrival of the maritime inflow and the embedded LLJ at around 2000 UTC, differential cooling occurs below around 500 m a.g.l., which is strongest between around 100 to 300 m a.g.l. (Fig. 10a) and reduces the static stability. At the same time wind speed in the residual

layer increases sharply with a LLJ axis near 250 m a.g.l., which increases the dynamically induced turbulence. Both processes lead to an abrupt decrease in Richardson number to values close to 0 (Fig. 10d). Simultaneously, near-surface TKE increases rapidly (Fig. 10c) and surface sensible heat flux decreases to values of around -20 W m$^{-2}$ (Fig. 10e). This marks the onset of the jet phase. The concurrent LLJ and high near-surface TKE values agrees with observations at Nangatchori in central Benin (Lothon et al., 2008). The jet phase lasts until LLC form at around 0330 UTC (Fig. 10a). After around midnight TKE decreases

(Fig. 10c) and stability slightly increases (Fig. 10d) which is likely related to an upward shift of the LLJ axis to around 400 m a.g.l. (Fig. (10a), which reduces the vertical wind shear below the LLJ axis. The measurements during IOP 15 show a relation between the Richardson number and near-surface TKE. This is also the case for the other IOPs: near-surface TKE is clearly related to the Flux-Richardson number as well as the Bulk-Richardson number (Fig. 11). High TKE values (larger than 0.3 m$^2$ s$^{-2}$) only occur when the Richardson numbers are below 0.1 indicating a regime where turbulence is dynamically generated.

The correlation between low Bulk-Richardson numbers and high TKE values indicates that turbulence is generated up to at least 200 m a.g.l.

The comparison of the temporal evolution of the potential temperature tendency profiles for IOP 15 (Fig. 10a) with the mean temperature tendency between late afternoon and LLC onset (Fig. 9a) suggests that a large part of the cooling below the LLJ axis happens during the jet phase (except for the shallow surface inversion which forms during the stable phase). To

estimate the heat budget terms, we use a time period within the jet phase confined by the first and last radiosounding. This requires at least two radiosoundings within the jet phase, which is not the case for IOP 4 and 14, excluding these IOPs from this analysis. As the cooling due to HADV with the maritime inflow already starts before the considered time period (Fig. 1), we cannot apply the assumption for HADV from Sect. 2.3.1 to estimate HADV for this period and only calculate TOT, RAD and TURB. As described in Sect. 2.3.3, we estimate TURB for the layer below the LLJ axis using surface sensible heat flux

values from both energy balance stations. TOT and RAD are vertically averaged up to the height of the LLJ axis. Each RAD and TURB explain about 22 % of TOT (Fig. 12). We speculate that a large part of the residuum during this period is related to horizontal advection with the maritime inflow. Furthermore, we expect vertical advection (which we cannot estimate) and horizontal advection related to temperature difference on a regional scale (which we also cannot estimate from the existing measurements) to contribute to TOT as well.



### 5.3 Trigger mechanisms of LLC

While horizontal cold air advection contributes significantly to the cooling of the ABL and is thus a necessary pre-condition for LLC (Sect. 5.1 and 5.2), the LLC themselves are not necessarily advected as seen in the satellite images (Sect. 3), but also form locally. We assess the possibility of three mechanisms triggering the LLC: during some nights, LLC form over higher terrain,

suggesting that orographic lifting constitutes the final trigger mechanism, which confirms the results of numerical simulations by Schuster et al. (2013) and Adler et al. (2017).

The second possible mechanism is related to shear in the nocturnal ABL. Zhu et al. (2001) study how the formation of shear-driven idealized clouds in the nocturnal ABL is related to land surface and ABL processes using a simple well-mixed boundary layer theory. In the presence of vertical wind shear, these authors divide the nocturnal ABL into three parts – a surface layer, a

mixed layer and a transition zone at the top of the nocturnal ABL. As long as the well-mixed conditions are met, a relationship between the nocturnal clouds and the lifting condensation level (LCL) calculated from surface data exists, i.e. the cloud base is around the LCL. Zhu et al. (2001) assume that the nocturnal clouds form due to processes related to the land surface and the ABL, when the LCL is lower than the nocturnal ABL height, i.e. when the LCL is within the mixed layer. We calculate the LCL from air temperature, $T$, and dew point temperature, $T_d$, measurements at 2 m a.g.l. with the formula LCL$= 125(T - T_d)$,

and compare it to the observed CBH. When calculating the LCL for the individual IOPs we find two distinct types: during IOPs 1, 5, 6 and 8 the CBH agrees pretty well with the LCL, while during the other IOPs the CBH is up to several 100 m higher than the LCL (examples for both types are shown in Figs. 13a and b). From radiosoundings performed during the stratus phase we calculate the Bulk-Richardson number for the sub-cloud layer as an indicator for turbulent mixing and find a relation between the stability in the sub-cloud layer and the difference between CBH and LCL (Fig. 13c). When the CBH coincides with the

LCL, Bulk-Richardson numbers are very small or even negative. This indicates that during nights when the CBH equals the LCL the sub-cloud layer is near-neutrally stratified and the sub-cloud layer is coupled to the surface, i.e. after Zhu et al. (2001) the LLC are triggered by shear-related ABL processes for these cases. Interestingly, all IOP nights with low CBH, i.e. below 130 m a.g.l. (Sect. 3), are also nights when the sub-cloud layer is coupled to the surface. One the other hand, when the CBH is larger than the LCL, there is no coupling and other processes must be responsible for the triggering of LLC.

The third mechanism relates to the expansion of LLC to the upstream side, i.e. when new LLC are triggered upstream of existing LLC. This upstream expansion of LLC also occurs in the numerical simulations by Adler et al. (2017). These authors identify a growth mechanism related to a modification in the stratification and LLJ profile in the model: in areas covered with LLC, maximum static stability and the LLJ axis occur in the upper part of the LLC layer and are thus shifted upwards compared to cloud-free areas (where the LLJ axis is near CBH). This results in horizontal convergence upstream of existing clouds and

enhanced upward motion which causes additional cooling and triggers new clouds. We find observational evidence for this mechanisms in the temperature and wind profiles from radiosoundings: the mean profiles reveal that static stability below and in the LLC decreases during the stratus phase compared to the jet phase (Figs. 7b, d), which we assume to represent the conditions in the cloud free areas. When looking at individual nights it is also evident that static stability increases around CTH (this feature is obscured in the mean profiles). The mean LLJ axis shifts upwards towards the top of layer with lower stability





during the stratus phase (Figs. 7a, d). The upward shift (often associated with a weakening) of the LLJ in the presence of LLC is nicely visible during IOP 3, 4, 8 and 15 in Fig. 3b, c, g and k. Even so we cannot directly measure the horizontal convergence upstream of LLC triggering new clouds, the observed impact of the LLC on stability and horizontal wind profiles strongly suggests that the mechanisms proposed by Adler et al. (2017) occurs in reality and may explain the upstream expansion of LLC.

Although we find observational evidence for all three possible mechanisms, their further analysis requires spatial information on the dynamic and thermodynamic conditions, which are not available from the observations at Savè, but could be investigated by combining the observations with numerical simulations.

## 6  Discussion

We discuss and relate the observed processes relevant for LLC formation with the processes postulated in former numerical studies. From the radiosonde data we estimate different terms of the heat budget and find that HADV with the maritime inflow accounts for about 50 % of the observed cooling at Savè (Sect. 5.2.1). Despite the uncertainty arising from the estimation of HADV (Sect. 2.3.1 and Figs. 4 and 9), we are confident that HADV actually depicts the largest contribution to TOT before LLC form. This confirms the model case study of Adler et al. (2017) who estimated heat budget terms from the 15-min model output for an area around Savè and the model study of Schuster et al. (2013) who calculated mean heat budget terms over 12 h periods (1800-0600 UTC) for cloudy and clear nights and averaged along a south-west to north-east oriented cross section. Schuster et al. (2013) found a cooling of around 5 K $(12\,\text{h})^{-1}$ with advection being the largest contribution at some distance from the coast. This value is similar to the overall potential temperature change between the stable phase and stratus phase (Fig. 7b). The absolute values of RAD derived with the radiative transfer model are the same order of magnitude as the modelled contribution by Schuster et al. (2013). We estimate TURB to cause about 22 % of TOT during the jet phase. The relevance of turbulent mixing below the LLJ axis is supported by profiles of radial velocity variance estimated from azimuthal Doppler lidar scans at 15 degree elevation angle, which show higher values in the shear layers above and below the LLJ axis (Babić et al., 2018). This confirms findings by Schrage and Fink (2012), Schuster et al. (2013) and Adler et al. (2017). Unfortunately, no general quantification of the turbulence is possible from the observations, although two Doppler lidars were deployed at Savè. It turned out to be difficult to obtain reliable measurements of vertical velocity fluctuations from the lidars during the night. In the nocturnal ABL over the southern great plains in the United states, where a strong LLJ develops during the night, Bonin et al. (2015) successfully used Doppler lidar measurements to relate the vertical velocity variance to the bulk wind shear caused by the LLJ. When wind speed exceeds a certain height-dependent threshold, strong bulk shear-generated turbulence develops. With wind speed values below this threshold no significant turbulence is detected by the Doppler lidars. At 200 m a.g.l., i.e. around the mean LLJ height at Savè, their threshold is around 15 m s$^{-1}$. As the LLJ at Savè hardly ever exceeds this value, this might explain why the Doppler lidars at Savè did not detect turbulence below the LLJ, although the weak stable stratification indicates that vertical mixing is present. We assume that the problem of the Doppler lidars to detect velocity fluctuations below a certain strength are related to the size of the pulse volume (i.e. range gate and pulse width) for which the




velocities are calculated. Velocity fluctuations within the pulse volume cannot be resolved and the estimated radial velocity is an average over the whole pulse volume. Thus, small scale vertical velocity fluctuations may exist but were not detected by the lidars.

Adler et al. (2017) hypothesize that gravity waves which form in the stably stratified layer of the LLJ axis contribute to a
cooling due to vertical cold air advection and lead to cloud formation in the wave crests. The existence of gravity waves could neither be verified nor disproved by the observations. On the one hand vertical velocity measurements with the Doppler lidars during the night were not reliable as described above and on the other hand no in situ aircraft measurements were conducted during the night.

From the radiosonde observations at Savè during IOP nights we find a strong relation between a decrease in temperature and
a very sudden increase in horizontal wind speed with an LLJ-shaped profile, which is why we attribute these changes both to the arrival of the maritime inflow. Note that this relation is less clear when using continuous remote sensing instruments and different objective criteria to detect the onset times (Dione et al., 2018). We propose therefore that the LLJ we observe at Savè is mainly linked to the maritime inflow and does not form locally. That means that the main reason for LLJ formation differs from those relevant for regions further in the north of southern West Africa, i.e. for the Sahel or Sahara (e.g. Lothon et al.,
2008). These authors attribute the formation of the LLJ to the relaxation of the friction force after sunset and to the temperature and pressure gradients linked to the Saharan heat low. This issue might be explained by the inland extent of the maritime inflow. Near the coast the maritime inflow dominates the ABL conditions before a local LLJ may form. For the regions further north, which are affected much later if at all by the maritime inflow, a LLJ may form locally related to the relaxation of friction force.

## 7   Summary and conclusions

Eleven IOP nights from the DACCIWA ground-based field campaign conducted in southern West Africa during the summer Monsoon season in 2016 are analyzed in order to characterize the spatial distribution and temporal evolution of LLC, to investigate the intra-night variability of ABL conditions and to assess the relevance of processes related to LLC formation. We used comprehensive observational data obtained at a supersite near Savè (Benin) from a ceilometer and cloud radar for the vertical extent of the LLC, UHF wind profiler and a Doppler lidar for wind information and energy balance stations for
the near-surface energy exchange. A large part of the analysis is based on radiosoundings perfomed in 1- to 1.5-h intervals providing temperature, humidity and wind profiles in the ABL.

Based on the dynamic and thermodynamic conditions in the ABL, different nocturnal phases are distinguished (Fig. 1): a surface inversion forms after sunset when the horizontal wind speed in the Monsoon layer is weak (stable phase). The jet phase starts with the arrival of the Gulf of Guinea maritime inflow, which propagates northwards during the evening. During this
phase a LLJ wind profile and differential horizontal cold air advection are prominent. Once saturation is reached LLC form either as stratus or stratus fractus at first. The key findings of this study are:

i. The vertical extent, coverage, onset time and horizontal distribution and evolution of LLC vary considerably for indi-
       vidual nights. The upper boundary of LLC ranges from 370 to 870 m a.g.l. and the lower boundary varies between 70





and 450 m a.g.l. During some nights, stratus forms very rapidly, while during other nights periods with stratus fractus precede stratus for several hours. LLC onset times are detected between 2100 and 0445 UTC. The location of the first formation of LLC as well as the following horizontal expansion differs: during some nights, LLC first occur over higher terrain, while during other nights, the location of the first formation seems to be independent of terrain features. LLC expand to the upstream as well as to the downstream side with respect to the mean south-westerly flow.

ii. The mean profiles for each of the nocturnal phases reveal some characteristic features: the surface inversion, which is present during the stable phase, gets eroded during the jet phase by differential cold air advection and shear-generated turbulence. During the stratus phase, static stability below and within the LLC layer decreases compared to the jet phase. A distinct LLJ profile is visible during the jet and stratus phases, with the LLJ axis being near CBH during the jet phase and shifting upwards during the stratus phase.

iii. We calculate the contributions of cooling and moistening to the relative humidity changes between the late afternoon and the LLC onset. The relative humidity increases by about 25 % near the surface and decreases linearly with height up to around 750 m a.g.l. This increase is for the most part caused by cooling. Specific humidity contributes to the relative humidity increase before and during the arrival of the maritime inflow at the site, but specific humidity decreases once Savè is within the maritime inflow. In total, moistening contributes very little to reach saturation in the ABL.

iv. We estimate different terms of the heat budget for the time period before the LLC onset from the observations to assess which processes contribute to cooling: the temperature tendency at Savè is quantified from radiosonde measurements; the contribution by radiative flux divergence is derived from a radiative transfer model; the contribution by horizontal advection with the maritime inflow is estimated from radiosonde measurements at the coast and at Savè; and the bulk contribution by sensible heat flux divergence below the LLJ axis is assessed based on the surface sensible heat flux measurements. Strong cooling occurs at Savè up to around 750 m a.g.l. with a large part being caused by horizontal advection. We vertically average the different contributions and find that around 50 % of the cooling before LLC formation is caused by horizontal cold air advection, while sensible heat flux divergence contributes about 22 % during the jet phase. Radiative flux divergence contributes roughly 20 % in the cloud-free nocturnal ABL – independent of the phase.

v. While horizontal advection of cool air with the maritime inflow is a necessary precondition for LLC formation, the LLC are not necessarily advected, but rather triggered by other mechanisms. Besides orographic lifting, we find evidence for two more possible mechanisms leading to LLC formation: during some nights, the sub-cloud layer is characterized by small Bulk-Richardson numbers and low static stability which indicates that the LLC are triggered by shear-related ABL processes. On the other hand, we find that the LLC impact the wind profile by shifting the LLJ axis upwards towards the cloud top – compared to cloud-free conditions. This supports the hypothesis of the model study by Adler et al. (2017) that horizontal convergence upstream of existing LLC due to the upward shift of the LLJ axis triggers new LLC.

By using observational data of 11 nights we are thus able to identify relevant processes for LLC formation, to quantify different terms of the heat budget, to identify typical dynamic and thermodynamic profiles for the different nocturnal phases and to





confirm hypotheses based on numerical simulations. The results can on the one hand serve for the validation of high-resolution or large-eddy simulations and on the other hand be used to identify flaws in global and climate models, which might add to the problems of these models to correctly simulate the LLC and the West African Monsoon system. Furthermore, this study adds to the development of a conceptual model explaining the evolution, maintenance and dissolution of the LLC, which will

5    be based on the observational and numerical analysis within the DACCIWA project.

*Data availability.*  After the DACCIWA embargo period, the data used in this study will be available on the SEDOO database (Derrien et al., 2016; Handwerker et al., 2016; Kohler et al., 2016; Wieser et al., 2016)

*Competing interests.*  The authors declare that they have no conflict of interest.

*Acknowledgements.*  The DACCIWA project has received funding from the European Union Seventh Framework Programme (FP7/2007-

10    2013) under grant agreement no. 603502. We also want to thank the staff of KIT (Karlsruhe Institute of Technology) and UPS (Université Toulouse) for helping to install and run the equipment as well as those from INRAB in Savè for allowing the equipment to be used on their grounds. Further we are grateful to Andreas Fink of KIT and his group for performing the radiosoundings at Accra.



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



**Table 1.** IOP numbers, corresponding dates and mean potential temperature, specific humidity, horizontal wind speed and wind direction in the lower 1000 m at the coast and Savè during the DACCIWA ground-based field campaign. Values are obtained from the radiosoundings in the late afternoon and averaged for all three coastal stations if available. IOPs highlighted in grey are utilized in this study.

| IOP | Dates | Potential temperature (K) Coast / Savè | Specific humidity (g kg$^{-1}$) Coast / Savè | Wind speed (m s$^{-1}$) Coast / Savè | Wind direction (degree) Coast / Savè |
|---|---|---|---|---|---|
| 1 | 17 - 18 June | - / 302.2 | - / 15.0 | - / 3.3 | - / 219 |
| 2 | 20 -21 June | 300.0 / 300.4 | 15.9 / 15.9 | 10.9 / 14.4 | 214 / 193 |
| 3 | 25 - 26 June | 299.7 / 302.7 | 16.1 / 17.2 | 7.8 / 2.8 | 228 / 176 |
| 4 | 28 - 29 June | 299.9 / 302.2 | 15.8 / 16.8 | 7.3 / 3.2 | 224 / 177 |
| 5 | 30 June - 1 July | 300.4 / 302.5 | 15.5 / 16.1 | 6.8 / 3.4 | 233 / 210 |
| 6 | 2 - 3 July | 300.0 / 303.5 | 16.0 / 17.1 | 3.9 / 2.1 | 183 / 116 |
| 7 | 4 - 5 July | 299.9 / 303.5 | 16.8 / 16.6 | 7.4 / 5.4 | 218 / 186 |
| 8 | 7 - 8 July | 300.4 / 303.4 | 16.8 / 16.8 | 7.2 / 0.9 | 235 / 259 |
| 9 | 10 - 11 July | 299.8 / 303.0 | 16.5 / 16.8 | 10.8 / 9.6 | 228 / 201 |
| 10 | 13 - 14 July | 299.9 / 301.4 | 14.9 / 13.6 | 5.5 / 3.7 | 250 / 301 |
| 11 | 17 - 18 July | 299.0 / 302.4 | 16.0 / 15.7 | 7.4 / 5.0 | 206 / 207 |
| 12 | 20 - 21 July | 299.7 / 301.0 | 14.8 / 16.8 | 7.8 / 5.6 | 248 / 246 |
| 13 | 23 - 24 July | 299.4 / 301.4 | 13.3 / 16.8 | 9.0 / 2.3 | 250 / 247 |
| 14 | 26 - 27 July | 298.4 / 301.7 | 14.8 / 15.6 | 4.9 / 2.9 | 232 / 172 |
| 15 | 29 - 30 July | 299.0 / 302.6 | 15.0 / 15.0 | 4.0 / 2.2 | 200 / 215 |



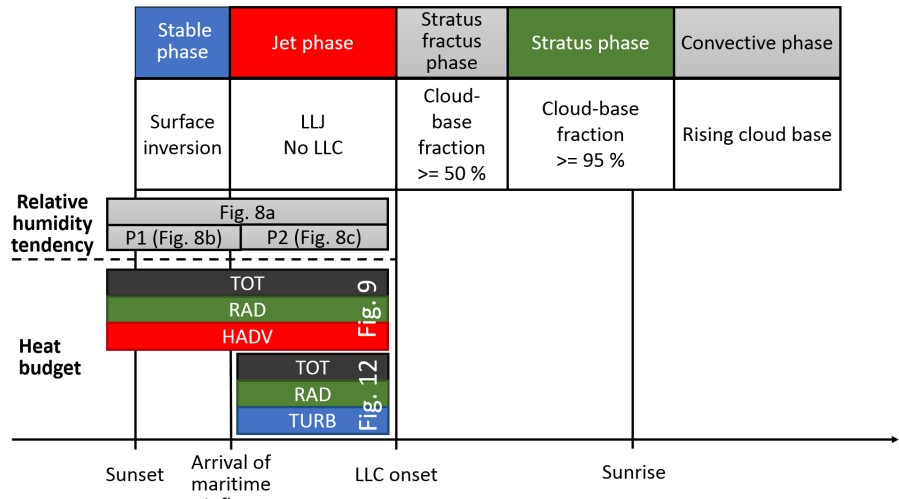

**Figure 1.** Schematic overview of the different phases including their main characteristics. Note that the stratus fractus does not occur during all nights. The different time periods considered for the estimation of the relative humidity changes and the heat budget terms are indicated.

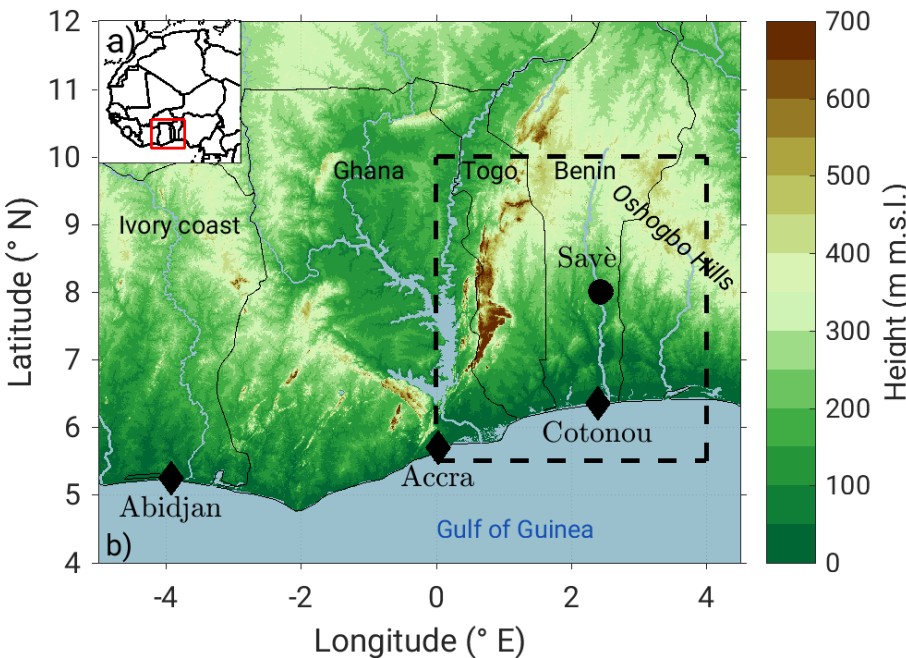

**Figure 2.** Location of the investigation area (red box) in southern West Africa (a) and topography in the investigation area with the location of the supersite Savè (circle) and of the coastal radiosonde stations at Abidjan, Accra and Cotonou (diamonds) indicated (b). The dashed box indicates the area used for the analysis of the spatial distribution of LLC. Solid lines indicate country borders with country names given in (b).



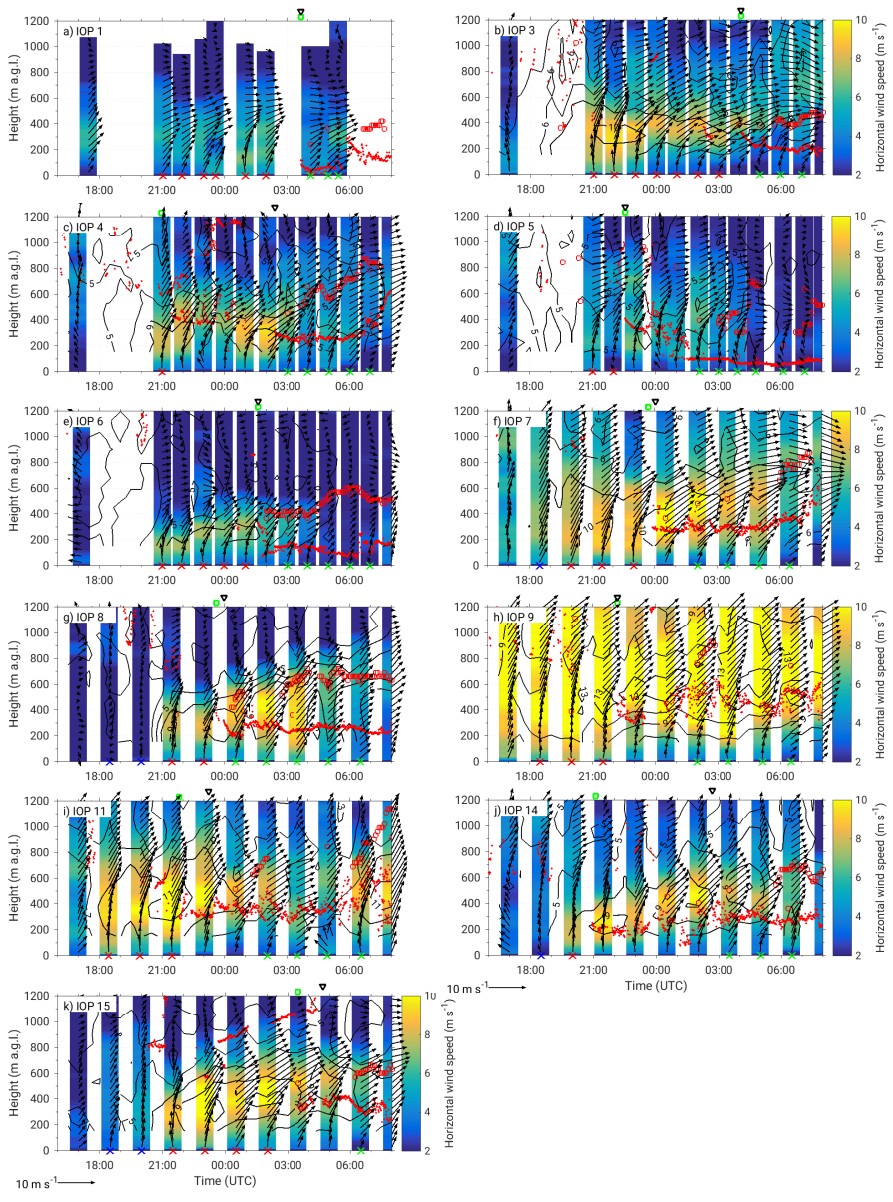

**Figure 3.** Horizontal wind speed from radiosondes (colour-coded) and from UHF profiler (contour, 2 m s$^{-1}$ increments) and horizontal wind vector from radiosondes (arrows) for IOPs analyzed in this study. The vertical bars are centered on the launch times of the radiosondes. Red dots indicate cloud base from ceilometer measurements and red circles cloud top from cloud radar measurements. Green and black markers at the top of each plot indicate the onset times of stratus fractus and stratus, respectively. Blue, red and green crosses at the bottom of each plot indicate radiosoundings considered in the averaged profiles for the stable, jet and stratus phases.




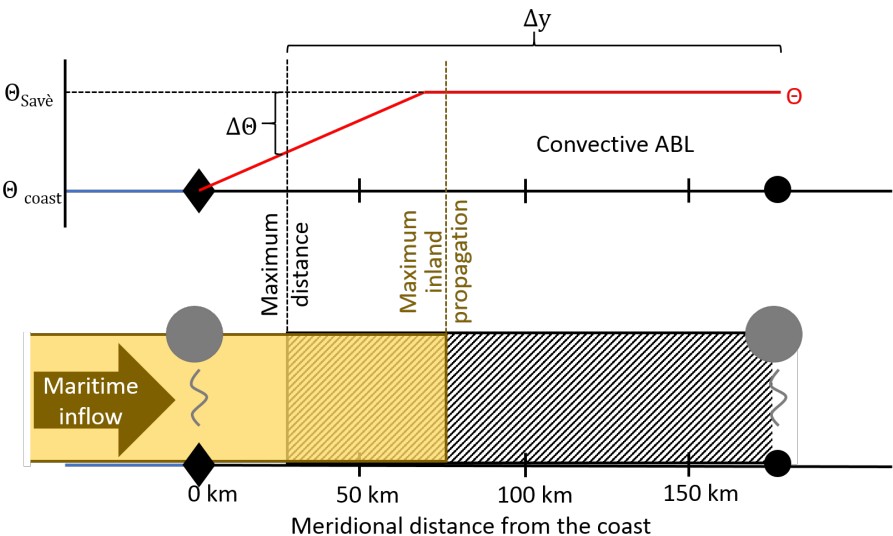

**Figure 4.** Schematic drawing illustrating the assumptions made for the estimation of horizontal advection using the example of a maximum inland propagation of the marine inflow of 75 km in the afternoon. In the bottom half the different air masses are indicated: the cool maritime inflow (yellow area) is propagating inland from the south. The propagation speed and the onset time of LLC determine the maximum distance from which the air mass reaching Savè before the LLC onset may originate (dashed area). Radiosoundings are performed at the coast (diamond) and at Savè (circle). In the upper half of the figure, the meridional potential temperature distribution, $\Theta$, is displayed. In the maritime inflow the temperature increases linearly northwards from the coast, while it is constant in the continental ABL. For the estimation of horizontal advection $\Delta\Theta$ and $\Delta y$ are used.



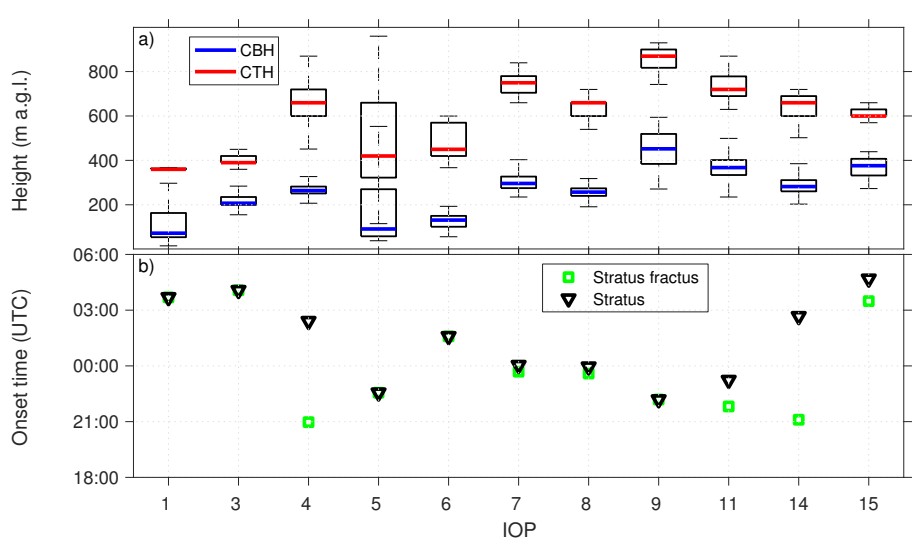

**Figure 5.** Cloud-base height (CBH) and cloud-top height (CTH) during the stratus phase (a). The blue and red bars indicate median CBH and CTH, respectively, the top and bottom edges of the boxes the 75th and 25th percentiles and the whiskers extreme values. Onset time of stratus fractus and stratus estimated from ceilometer measurements derived with the methods described in Sect. 2.2.2 (b).





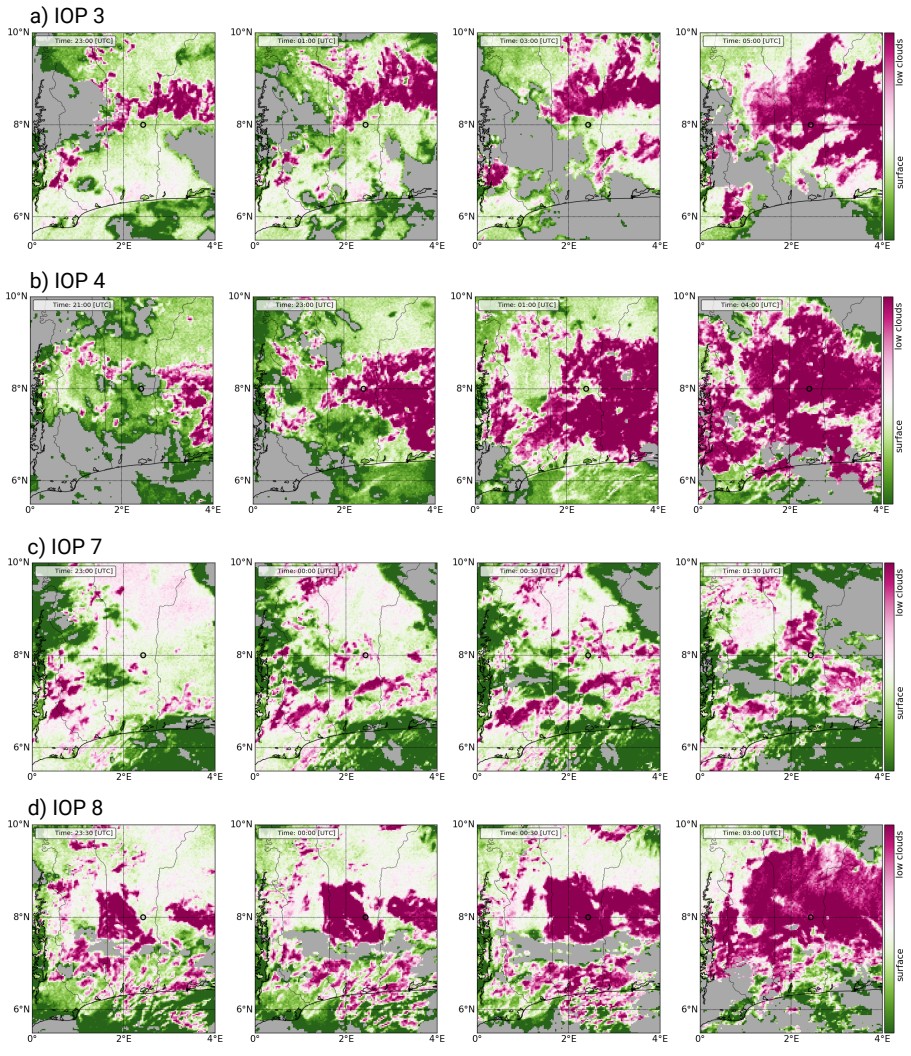

**Figure 6.** Brightness temperature difference of the thermal infrared channel at 10.8 and 3.9 μm for IOP 3 (a), 4 (b), 7 (c) and 8 (d). Purple color indicates LLC. The grey shading marks areas where the brightness temperature of the 10.8 μm channel exceeds 283 K, as an indicator for higher-level clouds. The maximum and minimum values of the color scale are 3.5 and -1.0, respectively.





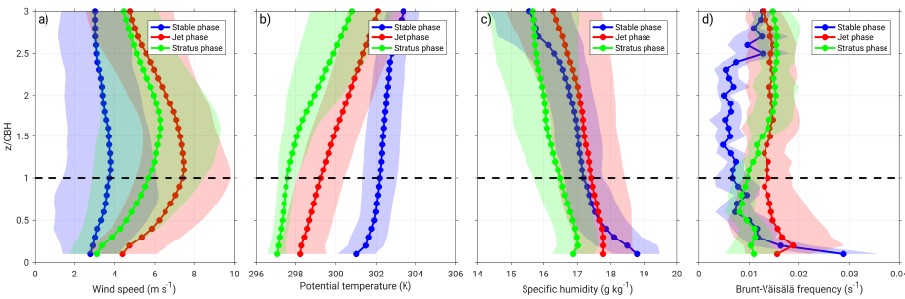

**Figure 7.** Mean profiles of horizontal wind speed (a), potential temperature (b), specific humidity (c) and Brunt-Väisälä-frequency (d) averaged for the stable phase, jet phase and stratus phase. The shading indicates the standard deviation. Before averaging the profiles are normalized with the median CBH for each IOP.

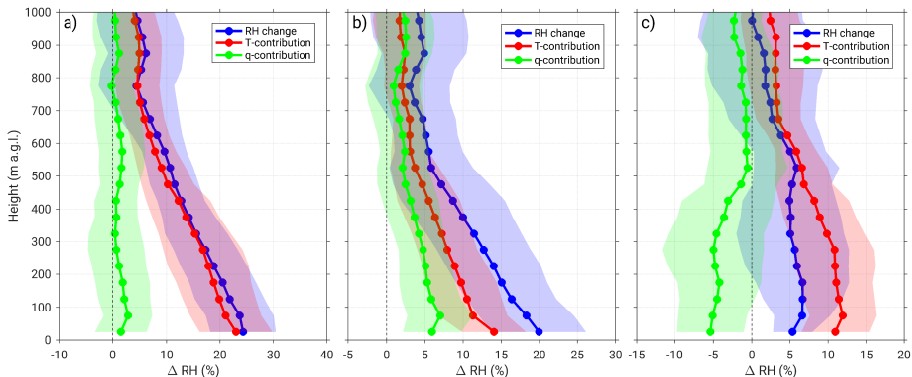

**Figure 8.** Mean profiles of relative humidity changes (RH change) and the contributions by temperature (T-contribution) and humidity (q-contribution) changes between 1700 UTC and the end of the jet phase (a), between 1700 UTC and the beginning of the jet phase (P1, b) and between the beginning and end of the jet phase (P2, c) averaged for all IOPs. For an overview of the considered time periods see Fig. 1. The shading indicates the standard deviation. Note that in (c), IOP 4 and 14 are not considered as not enough soundings exist for period P2.



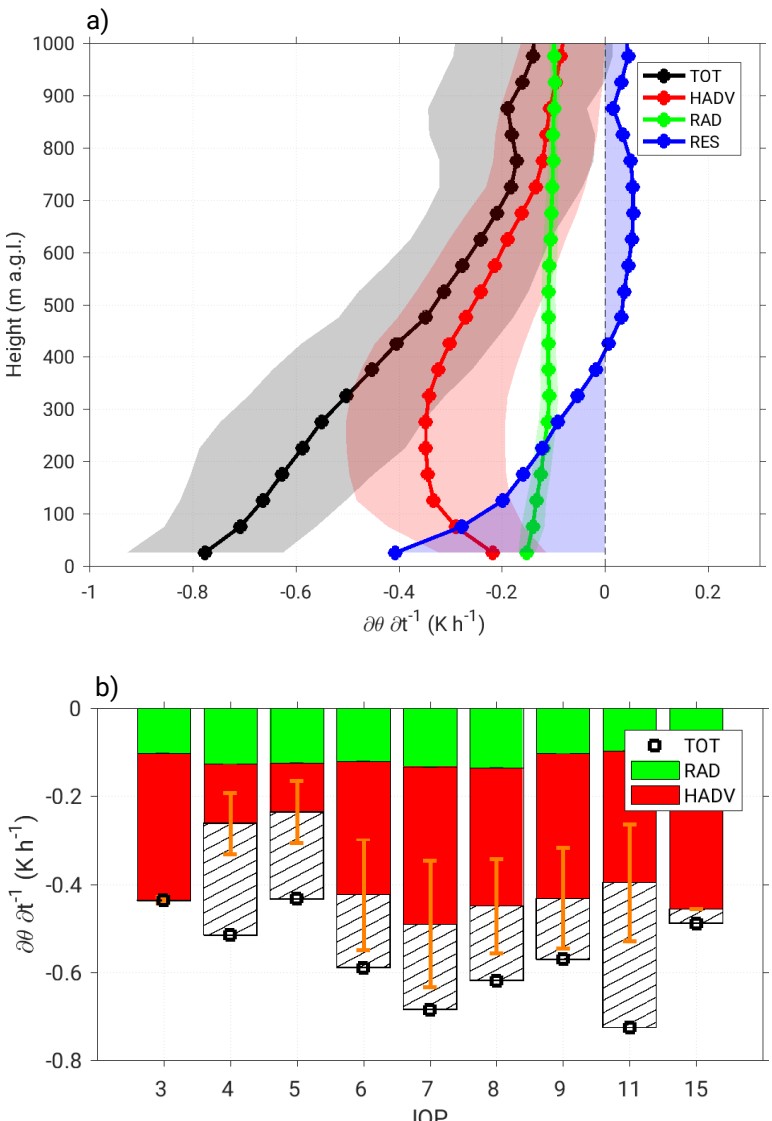

**Figure 9.** Mean profiles of potential temperature tendency (TOT), the contributions by horizontal advection (HADV) and radiative flux divergence (RAD) and the residuum (RES = TOT-HADV-RAD) between 1700 UTC and the end of the jet phase (see Fig. 1) averaged for all IOPs (a). Shading for TOT, HADV and RAD indicates the standard deviation and shading for RES marks the area between the RES and 0 K $h^{-1}$. Vertically averaged TOT, RAD and HADV for individual IOPs (b). The values are averaged up to the height where TOT decreases below 1/e of the maximum cooling. The dashed areas indicate the residuum. The errorbars for HADV result from the usage of three coastal stations and four maximum inland propagation distances of the maritime inflow to estimate HADV. Note that IOPs 1 and 14 are not considered due to missing soundings at the coast.





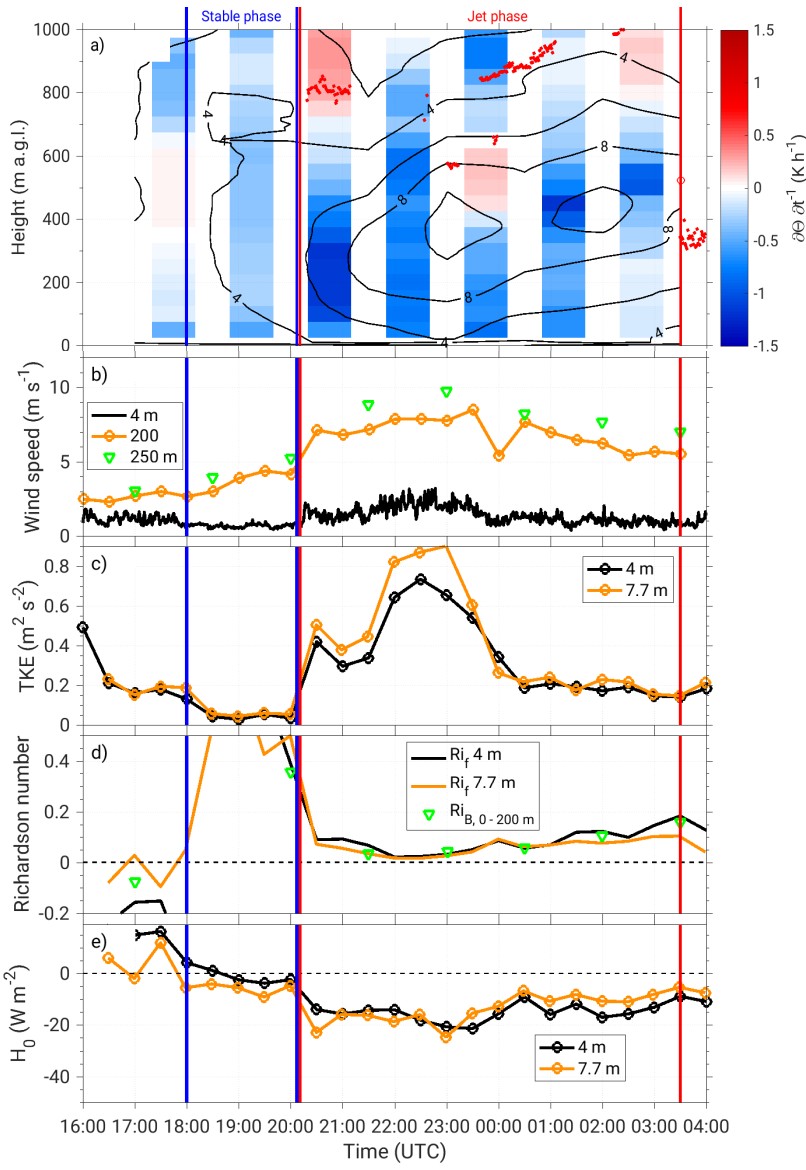

**Figure 10.** Time-height plot of potential temperature tendency (colour coded) and horizontal wind speed (contours) from radiosondes (a) and timeseries of horizontal wind speed from near-surface measurements at 4 m a.g.l., Doppler lidar at 200 m a.g.l. and radiosondes at 250 m a.g.l. (b), turbulent kinetic energy, TKE, (c), Flux-Richardson number, $Ri_f$, and Bulk-Richardson number between the surface and 200 m a.g.l., $Ri_{B,0-200\ m}$, (d) and surface sensible heat flux, $H_0$, (e) for IOP 15. TKE, $Ri_f$ and $H_0$ are measured at 4 and 7.7 m a.g.l. Blue and red vertical lines indicate the stable phase and jet phase, respectively. In (a), each bar represents a potential temperature profile calculated and centered in between consecutive soundings.





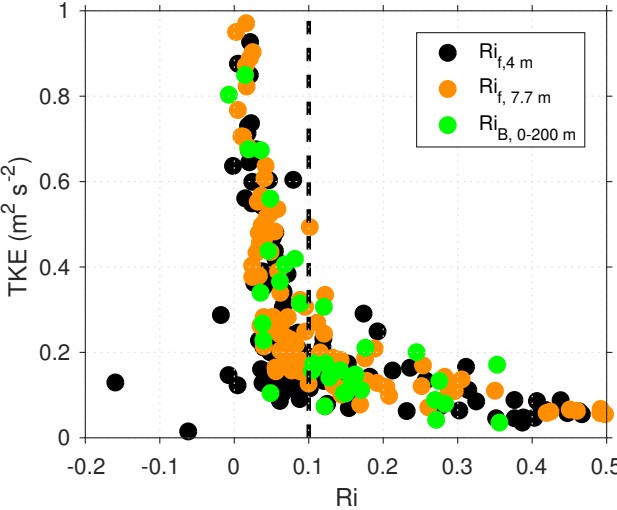

**Figure 11.** Relation between turbulent kinetic energy, TKE, and Richardson number, Ri, for the cloud-free time periods between 1830 and 0600 UTC during all IOPs. TKE is always measured at 4 m a.g.l., while Flux-Richardson numbers are measured at 4 m a.g.l., $Ri_{f,4\text{ m}}$, and 7.7 m a.g.l., $Ri_{f,7.7\text{ m}}$, and Bulk-Richardson number is calculated between the surface and 200 m a.g.l. using radiosonde data, $Ri_{B,0-200\text{ m}}$.

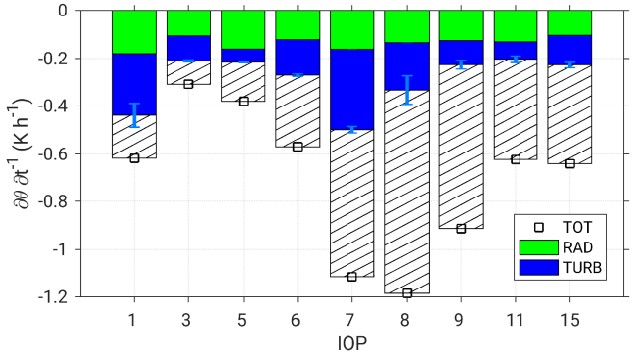

**Figure 12.** Vertically averaged temperature tendency (TOT), radiative flux divergence (RAD) and sensible heat flux divergence (TURB) for individual IOPs. The values are averaged up to the height of the LLJ axis and calculated for a period within the jet phase (Fig. 1). The dashed area indicates the amount of cooling which cannot be explained by neither RAD nor TURB. The errorbar for TURB results from the usage of the surface sensible heat fluxes from two energy balance stations.





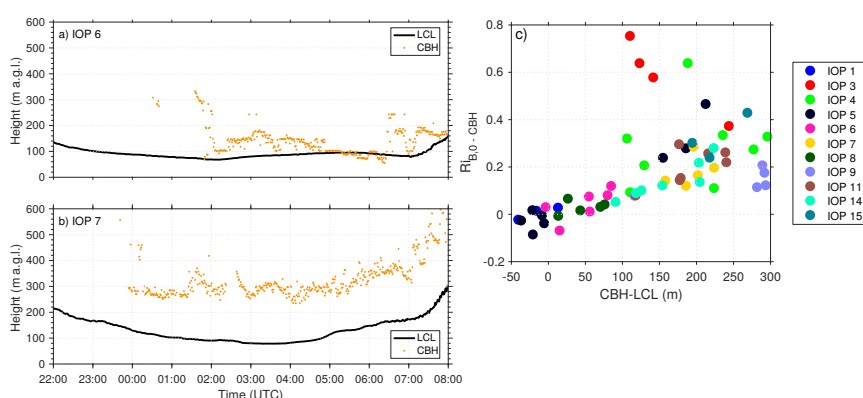

**Figure 13.** Cloud-base height (CBH) measured by ceilometer and lifting condensation level (LCL) calculated from surface measurements at 2 m a.g.l. for IOPs 6 (a) and 7 (b). Relation between the Bulk-Richardson number calculated from radiosoundings for the sub-cloud layer ($Ri_{B,0-\mathrm{CBH}}$) and the difference between CBH and LCL (c).