# Peer review of "Nocturnal low-level clouds in the atmospheric boundary layer over southern West Africa: an observation-based analysis of conditions and processes"

_Atmospheric Chemistry and Physics, 2018_

## Referee Comment (RC1) · Anonymous Referee #1 · 23 Sep 2018

Review of "Nocturnal low-level clouds in the atmospheric boundary layer over southern West Africa: an observation-based analysis of conditions and processes" by Adler, B., K. Babic, N. Kalthoff, F. lohou, M. Lothon, C. Dione, X. Pedruzo-Bagazgoitia, and H. Andersen. [Ref.: acp-2018-775]

This work characterizes nocturnal cloud formations (specifically stratus fractus and stratus) over a specific region of West Africa as observed during the ground based field project DACCIWA. The work essentially builds on studies already published by other co-authors on this paper to expand analyses to multiple IOP cases and to help

validate/understand modeled results.

This paper addresses a scientific topic within the scope of ACP, though it does not present a novel concept or approach since it essentially uses ideas and concepts from previous work. However, the fact that this region is particularly understudied and provides for a better understanding of cloud formation during the night in a monsoonal region, it is still of interest and value to the community.

The title accurately describes what they attempted to do in their experiment and the abstract summaries the work. They do give proper credit to previous work and use well known papers by in this particular sub field. The overall presentation is well structured and broken up into distinct sections to help make the work better to understand. There are some minor writing sytles writing style, grammar, typographical errors and other wording issues throughout the paper, but nothing that cannot be easily addressed. There accurately represent the mathematical formula for their heat budget. The figures can be improved, but again, nothing that is particularly difficult to be fixed.

This paper is publishable with minor revisions. Minor comments and suggestions are below. Specific comments are below.

MAJOR COMMENTS: 1. There could be more justification of why this region is of particular importance. In the conclusions section (page 16 line 32 through page 17 line 5) there is a good summary of this, but it is lacking in the introduction. 2. The figures (see specific comments below) can be improved to help the readers experience.

SPECIFIC FIGURE COMMENTS Figure 1) How long (on average) are these phases? Are they and hour? 5 hours? Are the sizes of the boxes in the figure related to how long each phase actually lasts? You can probably put a median and range for each phase in the caption. Does it vary from night to night by a lot? Figure 2) Are the three other marked areas that Ghana, and Nigeria locations mentioned on page 2? Abidjan is in the Ivory Coast not Nigeria. You don't list the countries here just the cities. In the caption identify Accra as the Ghana site, Abidjan as the Ivory Coast site and Cotonou

[Figure]

and Save as the Benin sites (the text for the country names blend in with the coloration for the topography and make them harder to read. Perhaps use white for the country names and black for the sites? Figure 3) This figure has the potential to very helpful to readers, however, everything is too small and the color shading is so dark it's hard to see the wind directions and other markings and contours. First, the x markers on the bottom of each panel are nearly impossible to see, they need to be bigger and bolded. Second, the shading should be lightened so you can see other things. Third, the triangles on top of each panel also need to be bigger, they're hard to see (partially because the green color doesn't show up well on a while background). In the caption you refer to the "stable, jet and stratus phases, which colors refer to these. You can put "stable (blue), jet (red), and stratus (green) to make this clear for the reader.

Figure 5) When the symbols (triangles and squares), does that mean that there was no stratus fractus?

Figure 6) These individual panels are small and hard to read, especially the text in the upper left corners of each panel for the times. It's impossible to read some of them. Since you have space for the IOP number perhaps put the time in the upper right hand corner ABOVE the panel in line with the IOP #. This way you can clearly see the time progression for each IOP. Where is Save on this map? Put a maker of some kind, perhaps in yellow or cyan that would stand out. Also, it would be interesting to see the flow directions on these maps. When you talk about the stratus expanding it would be a neat annotation to show the direction with an arrow of some kind to help readers visualize the changes.

Figure 9) In the caption "residuum" should be "residual"

Figure 10) For panel d) the Bulk Richardson number looks like it is going off of the axis. Why don't you just make the y-axis 0.6 or more so we can see how high it actually goes between 18:00 and 20:00, especially since you have the markers there for the stable phase.

Figure 11) In the caption "Relation" should be "Relationship"

Figure 13) Enlarge the dots for CBH. They are hard to see. In the caption "Relation" should be "Relationship"

SPECIFIC TABLE COMMENT Table 1) There are so many shaded and the shading may or may not work well when published. Perhaps shade the ones that you're NOT going to use? This way you're blocking them out and the ones you are using are unshaded.

MINOR COMMENTS: ABSTRACT Line 5: "decisive" is an odd word choice (you actually use it throughout) and I think you may mean "necessary" Line 6-7: "The aim is to study LLC" should be "The DACCIWA project studies LLC" Line 8: "Typical nocturnal phases" is unclear at this point, what do you mean by phase? Even in the abstract it should be clear what you are referring to. Line 12-14: What is the difference between "relative humidity" and "moisture" Line 13: "decisive" again would more appropriately be "necessary" Line 14: remove "the" before "the LLC" Line 16: remove "of" before "of LLC"

1. INTRODUCTION (page 2) Line 30-33: You list the locations for the ground stations but you don't mention "Savè" here (and then you do on the next page but don't specify it's in Benin. I suggest making a new Figure 1 that shows ALL of the ground sampling stations including Savè. Also, look at comments for Figure 2, Abidjan is in the Ivory Coast not Nigeria, what site is in Nigeria?

1. INTRODUCTION (page 3) Line 1: As per comment on the last page, you should specify that Save is in Benin, otherwise it is unclear at this point where your main location is. Line 13: "41-days" should be "41-day" Not plural. Line 18: "as well as" should be "and" Line 22: "as well as" should be "and"

2. DATA USED AND METHODS (page 3) Line 31: "found north and east of it." should be "found to the north and east."

2. DATA USED AND METHODS (page 4) Line 6: "as well as" should be "and" – NOTE: In general you should use "and" instead of "as well as" you use this phrase much too frequently. Line 16: What do you mean by the "nominal times?" It is unclear in the context of this sentence.

2. DATA USED AND METHODS (page 5) Line 2: Why is the threshold -35 dBz? Do you have a reference for this? Is this the standard threshold for clouds or are you doing something specific for you analysis that differs from a standard used in other works? Line 29: Wording: "illustrate the LLC during the night" – perhaps a better choice would be "visualize the LLC during the night"

2. DATA USED AND METHODS (page 7) Line 27: "have a by minimum" should be "have a minimum" Line 25: "as well as" should be "and"

3. LLC CHARACTERISTICS (page 8) Line 14: "allows to obtain" should be "allows us to obtain" Line 25: In regards to this comment "indicated by the unfilled green and black markers" and Figure 3, these markers are VERY tiny and almost impossible to see the way the figure is sized. Even if they are a bit larger for the final paper these markers should be enlarged.

3. LLC CHARACTERISTICS (page 9) Line 1: 80 minutes (the root mean square error) is quite a long time in terms of cloud development and cloud properties. Clouds change quite rapidly and the difference between methods (ceilometer and cloud-radar) could provide very different interpretations. Line 7: "form already at" should be "form already" Line 8-9: "...only little (Fig. 6a). After that, the LLC suddenly start o expand to the south-west until they cover..." should be "...only a little (Fig. 6a). After, the LLC suddenly expands by xx% to the southwest until covering..." Also, be more specific about the expansion/increase in area, is it 1%, 10%, 5%? Line 13: "allowing to detect" should be "allowing for the detection of" Line 16: "...CBH are then rather homogeneous as visible at Save" should be "...CBH are rather homogeneous and visible at Save" NOTE: This sentence is unclear when you read it the first time, consider changing the

way you are phrasing this to be more clear. Line 18: "grow in the subsequent" should be "grow during subsequent hours" Also, perhaps instead of "midnight" use 0000 so the time referencing is consistent. Line 21: "first LLC form" should be "LLC first form" Line 23: What latitudes are you referring to for the "roughly zonal band?" How wide is it?

5.1 RELATIVE HUMIDITY CHANGES (page 10) Line 22: What do you mean by "placement of the period?" It isn't clear what you are referring to here. Line 27: "humidity increase are related" should be "humidity increase is related" Also, what do you mean by "only a little?" Be specific! 1%, 5%? Line 30: "lower layer are caused" should be "lower layer is caused" Also, this sentence is confusing. It is unclear what the "contribution of about 25%" refers to. Clarify.

5.1 RELATIVE HUMIDITY CHANGES (page 11) Line 1: "decisive" should be "necessary"

5.2.1 HEAT BUDGET ESTIMATES ... (page 11) Line 19: "residuum" should probably be "residual" Usually, residuum is used in chemistry and "residual" is used for other things like math and physics.

5.2.2 SENSIBLE HEAT FLUX ... (page 12) Line 3: "illustrated in the following using the example" should be "illustrated using the example" Line 16: " IOP 15 show a relation" should be "IOP 15 show a relationship" Line 31: "residuum" should be "residual"

5.3 TRIGGER MECHANISMS OF LLC (page 13) Line 18: "find a relation between" should be "find a relationship between" Line 21: there should be a comma after Zhu et al. (2001) Line 31: "mechanisms" should be "mechanism" Line 35: "top of layer" should be "top of the layer"

6. DISCUSION (page 15) Line 9: "strong relation between" should be "strong relationship between" Line 10: "attribute these changes both to" should be "attribute both these changes to"

7. SUMMARY AND CONCLUSIONS (page 16) Line 30: "model study" should be "modeling study"

---

## Referee Comment (RC2) · Anonymous Referee #2 · 4 Oct 2018

Review of "Nocturnal low-level clouds in the atmospheric boundary layer over southern West Africa: an observation-based analysis of conditions and processes"

In this paper, the night-time formation of low level clouds over the West African Monsoon region, or to be more precise, over southern Benin, is analyzed based on observations made during a field campaign. The relative contribution of relevant processes is analyzed based on radiosonde, lidar, radar, and ground measurements. Measurements from this region are very rare, this alone would make the paper an interesting read. Furthermore, research questions directly related to current difficulties in the

modelling of the West African Monsoon system are addressed. The paper conforms findings from prior modeling studies. Accordingly, it does not necessarily provide new insights, but confirms previous work, which was not based on observational evidence. I explicitly welcome the publication of such studies. The manuscript fits well into the scope of ACP and is based on new data.

The manuscript is suggested for publication after the below listed concerns are addressed.

MAJOR COMMENTS: 1. There is a second manuscript from the same group of authors under review at ACP. In the other manuscript (acp-2018-776) one particular night is discussed in more detail, while in this manuscript statistics over 11 nights are presented. Methods and results show a significant overlap. This manuscript refers a lot to the other manuscript, almost every section contains something like "more details can be found in BabicÌĄ et al. 2018". Although the authors discuss their second paper briefly in the introduction, it is not immediately clear to the reader which research questions the other one does not answer and why a second one is necessary.

MINOR COMMENTS: 1. Page 4, Line 14-15: IPO 10 was not used, because no clouds did form during this night. It is okay to leave out a day if the conditions don't fit, but it would still be interesting to check the results on the basis of this day. How does it differ from the other days? Was the jet weaker? Any other differences? In the discussion, the omitted day could be addressed again. The findings may help to explain why clouds did not form.

2. Page 11, Line 2-3: "The small moisture changes indicate that the moisture content in the maritime air mass is roughly the same as in the continental ABL, i.e. no pronounced zonal moisture gradient prevails between Savè and the coast." This statement don't seems to be in agreement with "Once Savè is within the maritime inflow air mass, specific humidity decreases working against the cooling with respect to the relative humidity change". Fig 8c also suggests that the advected air is drier. What is the

reason if there is no moisture gradient between the coast and Savè?

3. Page 11, Line 22: The threshold of TOTmax e-1 looks a little arbitrary, where does it come from?

4. Page 13, Line 14: The calculation of LCL from surface values don't seems to be necessary in the presence of radiosondings. Please comment on the reason not to use the radiosondings for this purpose.

5. Page 15, Line 15: Reason for LLJ formation: In my opinion, the maritime inflow is a direct consequence of the relaxation of the friction force and the pressure gradient related to the Saharan heat low. From that point of view, I don't see a different mechanism at work. Please comment a bit more on the difference and on the driving force of the maritime inflow.

6. Figure 1: please extent the figure caption a bit to include the abbreviations. The figure is referred to in the text before the introduction of the balance equation.

7. Figure 6: The labels are hardly readable, which means that it does not become immediately clear that the development over time is shown.

TYPOS etc.: 1. Page 8, Line 8: "vertical" instead of "horizontal" profiles are meant, correct?

2. Page 10, Line 6: Is "z/CBH" a name of a variable? Something like "z subscript CBH"?

---

## Referee Comment (RC3) · Anonymous Referee #3 · 10 Oct 2018

Review: Adler et al. Nocturnal low-level clouds in the atmospheric boundary layer over southern West Africa: an observation-based analysis of conditions and processes.

This study uses plentiful surface-based and radiosonde-based data to study the onset of low cloud cover as it forms in Benin between the coast and inland areas. Data primarily come from profilers and radiosondes with satellite cloud data also used. The goal of the study is to decipher which mechanisms are primarily responsible for low cloud formation in the area. During the study period, it is shown that cloud cover generally follows the arrival of a jet of cool marine air surging northward from the coast

in the evening. The horizontal advection of cool air by this nocturnal jet is the primary driver of cloud formation, though sensible heat and radiational flux divergence also contribute to cooling. Changes in humidity are negligible in the period preceding cloud formation, suggesting that cooling by cold air advection, not moistening is the main process responsible for cloud formation.

The observational network used is well suited for this study and the methods describing the data are well explained. The study argues convincingly that northward cold-air advection associated with the nocturnal jet is the primary driver of cooling and cloud formation. Aside from a few small suggestions, I would recommend publishing with only very minor revisions.

Comments:

One thing that I would like to know is how representative this short observation period is compared to long-term averages. With less than 15 IOPs over 7 weeks, it is possible that we are observing anomalous conditions or an odd year/season. Can the authors put their observation period into climatological context, showing whether there are any outstanding or unique conditions present during the period? Conditions such as SST offshore, monsoon strength, wind or temperature anomalies. . .

The authors present numbers concerning the % of cooling associated with three different mechanisms. These numbers look reasonable, but it would be good to explain in a little more detail how they were calculated, and especially how much uncertainty there is. I don't get a good sense from the text about the error in the results.

There are some minor copy editing issues, but I'm sure the journal will address these without my nitpicking.

---

## Author Comment (AC1) · 23 Nov 2018

**Response to the referee comments (ACP-2018-775)**

**"Nocturnal low-level clouds in the atmospheric boundary layer over southern West Africa: an observation-based analysis of conditions and processes"**
**by Bianca Adler et al.**

November 23, 2018

We thank the three anonymous reviewers for their clear and helpful comments which certainly improved the manuscript. We respond to all comments of the reviewer in this document and we prepared a revised manuscripts where major changes are marked in red. In the following, comments of the reviewers are given in italic style and our responses are given in normal font style. The changes which we made to the manuscript are copied and pasted and highlighted in red.

**1 Anonymous Referee #1**

*This work characterizes nocturnal cloud formations (specifically stratus fractus and stratus) over a specific region of West Africa as observed during the ground based field project DACCIWA. The work essentially builds on studies already published by other co-authors on this paper to expand analyses to multiple IOP cases and to help validate/understand modeled results. This paper addresses a scientific topic within the scope of ACP, though it does not present a novel concept or approach since it essentially uses ideas and concepts from previous work. However, the fact that this region is particularly understudied and provides for a better understanding of cloud formation during the night in a monsoonal region, it is still of interest and value to the community. The title accurately describes what they attempted to do in their experiment and the abstract summaries the work. They do give proper credit to previous work and use well known papers by in this particular sub field. The overall presentation is well structured and broken up into distinct sections to help make the work better to understand. There are some minor writing sytles writing style, grammar, typographical errors and other wording issues throughout the paper, but nothing that cannot be easily addressed. There accurately represent the mathematical formula for their heat budget. The figures can be improved, but again, nothing that is particularly difficult to be fixed. This paper is publishable with minor revisions. Minor comments and suggestions are below. Specific comments are below.*

**MAJOR COMMENTS**

1. *There could be more justification of why this region is of particular importance. In the conclusions section (page 16 line 32 through page 17 line 5) there is a good summary of this, but it is lacking in the introduction.*

   **Response:** Thank you for this comment. We added two sentences to the introduction explaining why understanding the conditions and processes related to LLC formation is important.

   "Numerical weather prediction and climate models still struggle to correctly represent the West African monsoon (Hannak et al., 2017), which may be related to the erroneous representation of the LLC. A profound and accurate understanding of the processes relevant for the formation, maintenance and dissolution of the LLC might help to identify flaws in these models."

**SPECIFIC FIGURE COMMENTS**

2. *Figure 1) How long (on average) are these phases? Are they one hour? 5 hours? Are the sizes of the boxes in the figure related to how long each phase actually lasts? You can probably put a median and range for each phase in the caption. Does it vary from night to night by a lot?*

   **Response:** As we did not use continuous measurements in this study to detect the onset of stable and jet phase, we cannot determine the exact length of these phases. As described in Sect. 2.3.1 (p.6, l. 3), we manually allocate the radiosonde profiles to the stable and jet phase taking into account the dynamic and thermodynamic characteristics of the phases (i.e. surface inversion and LLJ). Using the radiosonde profiles we can only give rough estimates on the duration of the phases. For stratus fractus and stratus phase, we used the continuous ceilometer measurements to detect the onset times and it is thus possible to give exact values for their length. Using rough estimates for stable and jet phases and exact mean values for stratus fractus and stratus phase we relate the width of the boxes in former Fig. 1 (now Fig. 2) to the duration of the phases. For stratus fractus and stratus we additionally calculate the mean and standard deviation of the duration and include this information in Sect. 3.

   "Onset times and durations of both phases vary considerably for the individual IOPs. During some nights long periods with stratus fractus precede stratus (e.g. 5.5 h during IOPs 4 and 14), while no stratus fractus at all is detected during other nights (IOPs 1, 3, 5, 6 and 9). The duration of the stratus phase varies between 2.3 h (IOP 15) and 8.8 h (IOP 9), assuming that the transition of the stratus to the convective phase starts at 0700 UTC. On the average, the stratus fractus phase lasts for $1.3\pm2.1$ h and the stratus phase for $5.6\pm2.3$ h."

[Figure]

Figure 1: Schematic overview of the different phases including their main characteristics. The width of the boxes is a rough estimate of the mean duration of the individual phases. Note that the stratus fractus phase does not occur during all nights. The different time periods considered for the estimation of the relative humidity changes and the heat budget terms are indicated.

3. *Figure 2) Are the three other marked areas that Ghana, and Nigeria locations mentioned on page 2? Abidjan is in the Ivory Coast not Nigeria. You don't list the countries here just the cities. In the caption identify Accra as the Ghana site, Abidjan as the Ivory Coast site and Cotonou and Save as the Benin sites (the text for the country names blend in with the coloration for the topography and make them harder to read. Perhaps use white for the country names and black for the sites?*

   **Response:** The locations of the the two other ground-based supersites in Ghana and Nigeria are not indicated in Fig. 2, because they are not used in this study. The sites Abdijan, Accra and Cotonou which are indicated in Fig. 2 are the coastal radiosonde stations, as indicated in the caption. In order to make the country names more visible we enlarged the font size (changing the color did not enhance the visiblity). We added the information that Savè is the ground-based supersite in Benin to the introduction and the country names to the corresponding sites in the caption of Fig. 2:

   "Based on the observational data gathered at the ground-based supersite in Benin (Savè, location in Fig. 1), which is the supersite with the most comprehensive instrumentation, a series of analysis have been conducted on the LLC: ..."

   "Location of the investigation area (red box) in southern West Africa (a) and topography in the investigation area with the location of the supersite Savè (Benin) and of the coastal radiosonde stations at Abidjan (Ivory coast), Accra (Ghana) and Cotonou (Benin) indicated (b). The dashed box indicates the area used for the analysis of the spatial distribution of LLC. Solid lines indicate country borders with country names given in (b)."

4. *Figure 3) This figure has the potential to be very helpful to readers, however, everything is too small and the color shading is so dark it's hard to see the wind directions and other markings and contours. First, the x markers on the bottom of each panel are nearly impossible to see, they need to be bigger and bolded. Second, the shading should be lightened so you can see other things. Third, the triangles on top of each panel also need to be bigger, they're hard to see (partially because the green color doesn't show up well on a while background). In the caption you refer to the "stable, jet and stratus phases, which colors refer to these. You can put "stable (blue), jet (red), and stratus (green) to make this clear for the reader.*

   **Response:** Following the reviewer's suggestion we modified Fig. 3: we removed the dark blue from the colormap so that the arrows are better visible. The markers at the top and bottom indicating the stratus and stratus fractus onset and the radiosoundings during the different phases are much bigger now. It is already indicated in the caption that blue, red and green crosses indicate radiosoundings within the different phases. Stratus fractus onset is now only plotted during nights when it was not detected at the same time as stratus. We copied the new figure into our response:

[Figure]

Figure 2: Horizontal wind speed from radiosondes (colour-coded) and from UHF profiler (contour, $2 \text{ m s}^{-1}$ increments) and horizontal wind vector from radiosondes (arrows) for IOPs analyzed in this study. The vertical bars are centered on the launch times of the radiosondes. Red dots indicate cloud base from ceilometer measurements and red circles cloud top from cloud radar measurements. At the top of each plot the onset time of stratus is indicated by the green triangle. The grey triangle marks the onset time of stratus fractus, if it occurred during the respective night. Blue, red and green squares at the bottom of each plot indicate radiosoundings considered in the averaged profiles for the stable, jet and stratus phases.

5. *Figure 5) When the symbols (triangles and squares), does that mean that there was no stratus fractus?*

   **Response:** Yes, when the onset times of stratus fractus and stratus match, there was now stratus fractus. We now use the same markers (diamonds) for stratus and stratus fractus and changed the colors for stratus and stratus fractus to green and grey, respectively, to be in accordance with the colors used in former Fig. 1 (now Fig. 2). We also added this information to the caption:

   "Onset time of stratus and stratus fractus (if it occurred) estimated from ceilometer measurements derived with the methods described in Sect."

6. *Figure 6) These individual panels are small and hard to read, especially the text in the upper left corners of each panel for the times. It's impossible to read some of them. Since you have space for the IOP number perhaps put the time in the upper right hand corner ABOVE the panel in line with the IOP #. This way you can clearly see the time progression for each IOP. Where is Save on this map? Put a maker of some kind, perhaps in yellow or cyan that would stand out. Also, it would be interesting to see the flow directions on these maps. When you talk about the stratus expanding it would be a neat annotation to show the direction with an arrow of some kind to help readers visualize the changes.*

   **Response:** Following the reviewer's suggestions we enlarged the time indication of each suplot and put it in the right hand corner above the panels. Savè is indicated by the black circle roughly in the middle of the image. We mention this now in the caption. We prefer not to fill the marker as this would mask the clouds directly over Savè. We decided not to add arrows indicating the temporal expansion of the low-level clouds, as this would be very subjective (and not possible for some panels of e.g. IOP 7). We think that the expansion of low-level clouds described in the text can be retraced by comparing the low-level cloud patches in consecutive panels by eye.

[Figure]

Figure 3: Brightness temperature difference of the thermal infrared channel at 10.8 and 3.9 μm for IOP 3 (a), 4 (b), 7 (c) and 8 (d). Purple color indicates LLC. The grey shading marks areas where the brightness temperature of the 10.8 μm channel exceeds 283 K, as an indicator for higher-level clouds. The maximum and minimum values of the color scale are 3.5 and -1.0, respectively. The location of Savè is indicated by the black circle.

7. *Figure 9) In the caption "residuum" should be "residual"*

   **Reponse:** We changed residuum to residual in the caption.

8. *Figure 10) For panel d) the Bulk Richardson number looks like it is going off of the axis. Why don't you just make the y-axis 0.6 or more so we can see how high it actually goes between 18:00 and 20:00, especially since you have the markers there for the stable phase.*

   **Response:** We limit the y-axis to 0.5 as we want to focus on the range with low Richardon numbers during the jet phase. For high Richardson number (e.g. above the often used critical Richardson number of 0.25) the flow is dynamically stable, which is the case during the stable phase. During the jet phase the Richardson number drops to very small positive values which is related to dynamically induced turbulence. If we increased the range of the y-axis the small values would become less visible.

9. *Figure 11) In the caption "Relation" should be "Relationship"*

   **Response:** We changed "Relation" to "Relationship".

10. *Figure 13) Enlarge the dots for CBH. They are hard to see. In the caption "Relation" should be "Relationship"*

    **Response:** We enlarged the dots for CBH and changed "Relation" to "Relationship" in the caption. We also changed several other occurrences of "relation" to "relationship" in the manuscript.

**SPECIFIC TABLE COMMENTS**

11. *Table 1) There are so many shaded and the shading may or may not work well when published. Perhaps shade the ones that you're NOT going to use? This way you're blocking them out and the ones you are using are unshaded.*

    **Response:** We removed the shading and now write IOPs not utilized in this study in italic font.

**MINOR COMMENTS**

**ABSTRACT**

12. *Line 5: "decisive" is an odd word choice (you actually use it throughout) and I think you may mean "necessary". Line 13: "decisive" again would more appropriately be "necessary"*

    **Response:** We don't think that "necessary" is the suitable expression in this context. We actually want to stress that we analyse the crucial factors for LLC formation. Expressions

describing this could be "crucial", "critical" or "decisive" according to a English dictionary. Thinking about the word again we decided to change the expression "decisive" to "crucial" throughout the manuscript.

"A unique data set was gathered within the framework of the Dynamics-Aerosol-Chemistry-Cloud-Interactions in West Africa (DACCIWA) project, which allows, for the first time, for an observational analysis of the processes and parameters crucial for LLC formation."

"The analysis of the contributions to the relative humidity changes before the LLC formation reveals that cooling in the atmospheric boundary layer is crucial to reach saturation...."

"Independent of the considered period, cooling is crucial to achieve saturation in the nocturnal ABL. "

13. *Line 6-7: "The aim is to study LLC" should be "The DACCIWA project studies LLC"*

    **Response:** The DACCIWA project addresses many more objectives than the ones related to LLC occurrence. The project objectives cover chemistry and aerosol aspects, large-scale dynamics, precipitation and health. It would therefore not be correct to state "The DACCIWA project studies LLC".

14. *Line 8: "Typical nocturnal phases" is unclear at this point, what do you mean by phase? Even in the abstract it should be clear what you are referring to.*

    **Response:** We added the information to the abstract that the phases are distinguished based on dynamic and thermodynamic conditions.

    "Based on the dynamic and thermodynamic conditions in the atmospheric boundary layer we distinguish typical nocturnal phases and calculate mean profiles for the individual phases."

15. *Line 12-14: What is the difference between "relative humidity" and "moisture"*

    **Response:** With moisture we mean the atmospheric water vapour content. To estimate the contributions to the relative humidity change we use specific humidity. We changed the expression moisture to specific humidity throughout the manuscript to avoid confusion.

16. *Line 14: remove "the" before "the LLC"*

    **Response:** Done.

17. *Line 16: remove "of" before "of LLC"*

    **Response:** We cannot find the phrase "of LLC" in line 16 of the abstract.

**INTRODUCTION**

18. *(page 2) Line 30-33: You list the locations for the ground stations but you don't mention "Savè" here (and then you do on the next page but don't specify it's in Benin. I suggest making a new Figure 1 that shows ALL of the ground sampling stations including Sav. Also, look at comments for Figure 2, Abidjan is in the Ivory Coast not Nigeria, what site is in Nigeria?*

    **Response:** First, we describe the DACCIWA project and state that airborne and ground-based measurements at three supersites were available. As we do not use any data from the supersite in Ghana and Nigeria, we prefer not to indicate these sites in Fig. 1 (former Fig. 2). The sites in Ivory Coast, Ghana and Benin indicated in the figure are the coastal radiosonde stations and not the ground-based supersites. We now specify on page 3 that we use data gathered at the ground-based supersite in Benin, which is at Savè.

    "Based on the observational data gathered at the ground-based supersite in Benin (Savè, location in Fig. 1), which is the supersite with the most comprehensive instrumentation, a series of analysis have been conducted on the LLC:... "

19. *(page 3) Line 1: As per comment on the last page, you should specify that Save is in Benin, otherwise it is unclear at this point where your main location is.*

    **Response:** Done. See response to previous comment.

20. *Line 13: "41-days" should be "41-day" Not plural.*

    **Response:** Done.

21. *Line 18: "as well as" should be "and" Line 22: "as well as" should be "and"*

    **Response:** Done.

**DATA USED AND METHODS**

22. *(page 3) Line 31: "found north and east of it." should be "found to the north and east."*

    **Response:** Done.

23. *(page 4) Line 6: "as well as" should be "and" NOTE: In general you should use "and" instead of "as well as" you use this phrase much too frequently.*

    **Response:** We thank the reviewer for this comment and replaced most of the "as well as" in the manuscript with "and".

24. *Line 16: What do you mean by the "nominal times?" It is unclear in the context of this*

*sentence.*

**Response:** With "nominal times" we refer to the standard radiosonde times, usually 00, 06, 12 and 18 UTC, used at operational radiosonde stations. We added the time information to the sentence to make it more clear.

"During IOPs, radiosondes were released at Savè one hour before the nominal times at 0000, 0600, 1200 and 1800 UTC. "

25. *(page 5) Line 2: Why is the threshold -35 dBz? Do you have a reference for this? Is this the standard threshold for clouds or are you doing something specific for you analysis that differs from a standard used in other works?*

**Response:** The threshold of -35 dBz is obtained manually by visual inspection of the plots of the radar reflectivity of hydrometeors from the cloud radar. The radar reflectivity gradients at cloud top are sharp and the detected cloud top height is not very sensitive to the threshold. We added this information to the manuscript. An example of radar reflectivity of hydrometeors is given in Babić et al. (2018), which nicely illustrates to sharp gradients.

"As the reflectivity changes abruptly near CTH, the determined CTH are not sensitive to the applied threshold value. "

26. *Line 29: Wording: "illustrate the LLC during the night" perhaps a better choice would be "visualize the LLC during the night"*

**Response:** We changed the text accordingly:

"In the absence of high or mid-level clouds, which are obscuring the LLC from the view of the satellite-borne sensor, the brightness temperature difference of the thermal-infrared channel at 10.8 µm and the middle-infrared channel at 3.9 µm are used to visualize the LLC during the night. "

27. *(page 7) Line 27: "have a by minimum" should be "have a minimum". Line 25: "as well as" should be "and"*

**Response: Done.**

**LLC CHARACTERISTICS**

28. *(page 8) Line 14: "allows to obtain" should be "allows us to obtain"*

**Response: Done.**

29. *Line 25: In regards to this comment "indicated by the unfilled green and black markers" and Figure 3, these markers are VERY tiny and almost impossible to see the way the figure is sized. Even if they are a bit larger for the final paper these markers should be enlarged.*

**Response:** We enlarged and filled the markers for the onset of stratus and stratus fractus in Fig. 3. See Fig. 2 in this response.

30. *(page 9) Line 1: 80 minutes (the root mean square error) is quite a long time in terms of cloud development and cloud properties. Clouds change quite rapidly and the difference between methods (ceilometer and cloud-radar) could provide very different interpretations.*

**Response:** We agree that the difference in onset times derived from different instruments (ceilometer and infrared camera) is substantial. As argued in the manuscript, this difference is caused by the methods, e.g. the onset times derived with the cloud camera strongly depends on the homogeneity of CBH, cloud-base cover and cloud density. Using the onset times from the cloud camera for the analysis could affect the mean profiles calculated for the stratus phase as some radiosonde profile would be additionally considered or removed from the stratus phase. However, as the radiosoundings were performed in 1-1.5 hourly intervals, the different onset times do not carry too much weight, especially considering the total number of 52 soundings during the stratus phase.

31. *Line 7: "form already at" should be "form already"*

**Response:** We do not understand this comment. In the manuscript it says "During IOP 3 LLC form already at 2200 UTC over ..." Removing the "at" does not make sense.

32. *Line 8-9: "...only little (Fig. 6a). After that, the LLC suddenly start to expand to the southwest until they cover..." should be "....only a little (Fig. 6a). After, the LLC suddenly expands by xx% to the southwest until covering..." Also, be more specific about the expansion/increase in area, is it 1%, 10%, 5%?*

**Response:** We use the satellite images simply to visualize the LLC evolution. For calculating expansion rates it would be necessary to define a mask applying objective criteria to detect pixels with LLC. We think that the additional required effort would not justify the added value and therefore we prefer to stick to the graphical representation.

33. *Line 13: "allowing to detect" should be "allowing for the detection of"*

**Response:** Done.

34. *Line 16: ". . .CBH are then rather homogeneous as visible at Save" should be ". . .CBH are rather homogeneous and visible at Save" NOTE: This sentence is unclear when you read it the first time, consider changing the way you are phrasing this to be more clear.*

**Response:** Thank you, this sentence was indeed not clear. We intend to say that the CBH are rather homogenous, which we know from the ceilometer measurements at Savè. We rephrased it:

"During this period, the CBH are rather homogeneous, which is visible in the ceilometer measurements at Savè (Fig. 3c)."

35. *Line 18: "grow in the subsequent" should be "grow during subsequent hours" Also, perhaps instead of "midnight" use 0000 so the time referencing is consistent.*

   **Response:** Done.

   "Then, LLC start forming at several locations in the domain and grow during subsequent hours, also occuring at Savè after 0000 UTC"

36. *Line 21: "first LLC form" should be "LLC first form"*

   **Response:** Done

37. *Line 23: What latitudes are you referring to for the "roughly zonal band?" How wide is it?*

   **Response:** We rephrased this sentence and "roughly zontal band" does not occur anymore:

   "Subsequently, the LLC expand in all directions and they cover most of the domain at sunrise."

**RELATIVE HUMIDITY CHANGES**

38. *(page 10) Line 22: What do you mean by "placement of the period?" It isn't clear what you are referring to here.*

   **Response:** With "placement of the period" we intend to refer to the time interval for which the changes are calculated. We rephrased the sentence:

   "...and the time interval for which the changes are calculated is indicated in Fig. 2)."

39. *Line 27: "humidity increase are related" should be "humidity increase is related" Also, what do you mean by "only a little?" Be specific! 1%, 5%?*

   **Response:** We modified the sentence:

   "Nearly 100 % of the relative humidity increase is related to cooling, while specific humidity changes contribute only little ($< 3$ %, Fig. 8a). "

40. *Line 30: "lower layer are caused" should be "lower layer is caused" Also, this sentence is confusing. It is unclear what the "contribution of about 25%" refers to. Clarify.*

   **Response:** We modified the sentence in order to clarify what we mean by 25%.

   "During P1, 5 % of the relative humidity increase (which is about one quarter of the total increase) in the lower layer is caused by an increase of specific humidity (Fig. 8b), while a decrease of specific humidity lowers the relative humidity increase by about 5 % during P2 (Fig. 8c)."

41. *(page 11) Line 1: "decisive" should be "necessary"*

    **Response:** As explained above we replace "decisive" with "crucial" as we think that "necessary" is not suitable.

**HEAT BUDGET ESTIMATES**

42. *(page 11) Line 19: "residuum" should probably be "residual" Usually, residuum is used in chemistry and "residual" is used for other things like math and physics.*

    **Response:** Done.

**SENSIBLE HEAT FLUX**

43. *(page12) Line 3: "illustrated in the following using the example" should be "illustrated using the example" Line 16: " IOP 15 show a relation" should be "IOP 15 show a relationship" Line 31: "residuum" should be "residual"*

    **Response:** Done.

**TRIGGER MECHANISMS OF LLC**

44. *(page 13) Line 18: "find a relation between" should be "find a relationship between" Line 21: there should be a comma after Zhu et al. (2001) Line 31: "mechanisms" should be "mechanism" Line 35: "top of layer" should be "top of the layer"*

    **Response:** Done.

**DISCUSSION**

45. *(page 15) Line 9: "strong relation between" should be "strong relationship between" Line 10: "attribute these changes both to" should be "attribute both these changes to"*

    **Response:** Done.

**SUMMARY AND CONCLUSIONS**

46. *(page 16) Line 30: "model study" should be "modeling study"*

    **Response:** Done.

**2 Anonymous Referee #2**

*Review of "Nocturnal low-level clouds in the atmospheric boundary layer over southern West Africa: an observation-based analysis of conditions and processes" In this paper, the night-time formation of low level clouds over the West African Monsoon region, or to be more precise, over southern Benin, is analyzed based on observations made during a field campaign. The relative contribution of relevant processes is analyzed based on radiosonde, lidar, radar, and ground measurements. Measurements from this region are very rare, this alone would make the paper an interesting read. Furthermore, research questions directly related to current difficulties in the modelling of the West African Monsoon system are addressed. The paper conforms findings from prior modeling studies. Accordingly, it does not necessarily provide new insights, but confirms previous work, which was not based on observational evidence. I explicitly welcome the publication of such studies. The manuscript fits well into the scope of ACP and is based on new data. The manuscript is suggested for publication after the below listed concerns are addressed.*

**MAJOR COMMENTS**

1. *There is a second manuscript from the same group of authors under review at ACP. In the other manuscript (acp-2018-776) one particular night is discussed in more detail, while in this manuscript statistics over 11 nights are presented. Methods and results show a significant overlap. This manuscript refers a lot to the other manuscript, almost every section contains something like "more details can be found in Babić et al. (2018)". Although the authors discuss their second paper briefly in the introduction, it is not immediately clear to the reader which research questions the other one does not answer and why a second one is necessary.*

   **Response:** Babić et al. (2018) present a detailed analysis of the diurnal cycle fo LLC for a case study of a typical night. They aim at identifying the main factors leading to the LLC formation and assess heat budget terms for different phases during the diurnal of cycle of the LLC. The aims of their paper are already outlined in the introduction (p. 3, l. 5-18). Because their results are based on a single case study, no conclusions on the representativeness of the results can be drawn. This is the reason for the present paper using data from 11 IOPs. By applying the same methods as Babić et al. (2018) we are able to generalize their findings and to address the representativeness of the results from the case study. We modified the introduction in order to clarify this point.

   "While the studies of Babić et al. (2018) and Dione et al. (2018) either look at the diurnal cycle during one case study or at mean quantities during a longer measurement period, the present analysis focuses on 11 IOP nights. As radiosoundings were performed in short temporal intervals of 1 to 1.5 h throughout the IOP nights, high-quality profile information on temperature, specific humidity and horizontal wind are available, which allows us to perform an analysis for these nights in a manner consistent with the methods used by Babić et al. (2018). By analyzing several nights – instead of one – we are able to address the representativeness of the results obtained from the single case study and to generalize some of the findings. Besides the generalization of process relevance for the formation of LLC, we aim to characterize the LLC and to investigate the intra-night variability of ABL conditions."

**MINOR COMMENTS**

2. *Page 4, Line 14-15: IOP 10 was not used, because no clouds did form during this night. It is okay to leave out a day if the conditions don't fit, but it would still be interesting to check the results on the basis of this day. How does it differ from the other days? Was the jet weaker? Any other differences? In the discussion, the omitted day could be addressed again. The findings may help to explain why clouds did not form.*

   **Response:** This is a very good point. The question why and under which conditions LLC form or do not form is very important. Because of this we are currently working on another paper dealing with this question, which will be submitted soon. Therefore, we decided not to include a comparison with LLC free nights in the present manuscript.

3. *Page 11, Line 2-3: "The small moisture changes indicate that the moisture content in the maritime air mass is roughly the same as in the continental ABL, i.e. no pronounced zonal moisture gradient prevails between Savè and the coast." This statement don't seems to be in agreement with "Once Savè is within the maritime inflow air mass, specific humidity decreases working against the cooling with respect to the relative humidity change". Fig 8c also suggests that the advected air is drier. What is the reason if there is no moisture gradient between the coast and Savè?*

   **Response:** This was misleading. We intended to say that specific humidity changes were overall small and their was no strong signal when looking at the whole time interval. For the shorter time periods (P1 and P2) specific humidity changes contributed to an increase and decrease of relative humidity, respectively. In order to avoid misunderstandings, we removed the sentence "The small moisture changes indicate that the moisture content in the maritime air mass is roughly the same as in the continental ABL, i.e. no pronounced zonal moisture gradient prevails between Savè and the coast." Instead we say:

   "The fact that specific humidity changes play a minor role might come a little unexpected. This is likely related to the relatively low sea surface temperature of the Gulf of Guinea limiting the specific humidity in the maritime ABL and high evapotranspiration from the dense vegetation over land leading to high specific humidity in the continental ABL. "

4. *Page 11, Line 22: The threshold of TOTmax e-1 looks a little arbitrary, where does it come from?*

   **Response:** We use the factor $e^{-1}$ to determine an objective threshold up to which we vertically average the temperature changes. The factor is based on the decay law ($N = N_0 \cdot e^{-\lambda t}$). A factor of $e^{-1}$ means that that the change is still 36 % (1/2.71) of a certain value

(in our case the maximum of TOT). Another option would have been to use 10 % of $\mathrm{TOT_{max}}$ as a threshold, which would correspond to $e^{-\lambda t} = 0.1$.

5. *Page 13, Line 14: The calculation of LCL from surface values don't seems to be necessary in the presence of radiosondings. Please comment on the reason not to use the radiosondings for this purpose.*

   **Response:** Radiosoundings have the disadvantage of being available only at certain times. This is why we use surface values to calculate LCL as they are continuous.

6. *Page 15, Line 15: Reason for LLJ formation: In my opinion, the maritime inflow is a direct consequence of the relaxation of the friction force and the pressure gradient related to the Saharan heat low. From that point of view, I don't see a different mechanism at work. Please comment a bit more on the difference and on the driving force of the maritime inflow.*

   **Response:** Thank you for this comment. This was not clear in the manuscript. As outlined in the introduction we assume that during the day a convergence zone is located along the coasts which separates cool maritime air from the warmer air in the convective ABL over land. Once turbulence decays in the convective ABL in the afternoon the cool maritime air is transported northwards with the monsoon flow, which is what we call the maritime inflow. Within the maritime inflow, relaxation of friction force leads to the formation of the LLJ-shaped wind profile which we observe. We think that the difference to the LLJ forming further in the north of southern West Africa is that in that region the LLJ forms locally and not within an advected air mass. In the region investigated in this study the conditions are affected by the maritime inflow before a local LLJ might form. We rewrote the paragraph in the discussion section to clarify this:

   "We assume that the relaxation of friction force leads to the formation of the LLJ-shaped profile within the maritime inflow. As the maritime inflow generally dominates the ABL conditions a few hours after sunset, there is not enough time for a pronounced LLJ to form locally without the influence of the maritime inflow. Based on the observational evidence, we propose therefore that the LLJ we observe at Savè is mainly linked to the maritime inflow and does not form locally. That means that the circumstances leading to LLJ formation differ from those relevant for regions further in the north of southern West Africa, i.e. for the Sahel or Sahara (e.g. Lothon et al., 2008). As the regions further north are affected much later if at all by the maritime inflow, the LLJ forms locally and is not linked to the maritime inflow. "

7. *Figure 1: please extent the figure caption a bit to include the abbreviations. The figure is referred to in the text before the introduction of the balance equation.*

   **Response:** We added a box with the abbreviations to the figure.

8. *Figure 6: The labels are hardly readable, which means that it does not become immediately clear that the development over time is shown.*

   **Response:** We enlarged the time label and put them outside the panels for better readability.

Reviewer 1 had the same comment.

9. *Page 8, Line 8: "vertical" instead of "horizontal" profiles are meant, correct?*

   **Response:** No, we mean profiles of horizontal wind. We changed the text to "vertical profiles of horizontal wind" in order to make it clear.

10. *Page 10, Line 6: Is "z/CBH" a name of a variable? Something like "z subscript CBH"?*

    **Response:** "z/CBH" depicts the normalisation of z with CBH. We added this explanation to the text:

    "To take into account the large variability of CBH during the individual IOPs (Fig. 5a), we normalize the profiles with the median CBH of each IOP (z/CBH). "

**3 Anonymous Referee #3**

*This study uses plentiful surface-based and radiosonde-based data to study the onset of low cloud cover as it forms in Benin between the coast and inland areas. Data primarily come from profilers and radiosondes with satellite cloud data also used. The goal of the study is to decipher which mechanisms are primarily responsible for low cloud formation in the area. During the study period, it is shown that cloud cover generally follows the arrival of a jet of cool marine air surging northward from the coast in the evening. The horizontal advection of cool air by this nocturnal jet is the primary driver of cloud formation, though sensible heat and radiational flux divergence also contribute to cooling. Changes in humidity are negligible in the period preceding cloud formation, suggesting that cooling by cold air advection, not moistening is the main process responsible for cloud formation. The observational network used is well suited for this study and the methods describing the data are well explained. The study argues convincingly that northward cold-air advection associated with the nocturnal jet is the primary driver of cooling and cloud formation. Aside from a few small suggestions, I would recommend publishing with only very minor revisions.*

**4 comments**

1. *One thing that I would like to know is how representative this short observation period is compared to long-term averages. With less than 15 IOPs over 7 weeks, it is possible that we are observing anomalous conditions or an odd year/season. Can the authors put their observation period into climatological context, showing whether there are any outstanding or unique conditions present during the period? Conditions such as SST offshore, monsoon strength, wind or temperature anomalies. . .*

   **Response:** Knippertz et al. (2017) present a detailed description of the large-scale setting of the DACCIWA field campaign. They find that the monsoon season 2016 was characterized

by Pacific La Niña and Atlantic El Niño and overall average rainfall. These authors also distinguished 4 distinct phases of the monsoon during the field campaign period. The analyzed IOPs are distributed over 3 of these phases: one IOP was during the pre-onset phase and the other 10 IOPs during phases when the monsoon conditions were relatively undisturbed and the strength and position of the Saharan heat low and African easterly jet were close to the climatological conditions. We added this information to the introduction and section 2:

"With respect to climatological conditions, the period of the field campaign was characterized by Pacific La Niña and Atlantic El Niño and overall average rainfall across the whole of West Africa (Knippertz et al., 2017). "

"IOP 1 falls into the pre-onset phase of the monoon and the other ten IOPs into phases during which the monsoon conditions are relatively undisturbed and the strength and position of the Saharan heat low and African easterly jet are close to the climatological average (Knippertz et al., 2017)."

2. *The authors present numbers concerning the % of cooling associated with three different mechanisms. These numbers look reasonable, but it would be good to explain in a little more detail how they were calculated, and especially how much uncertainty there is. I don't get a good sense from the text about the error in the results.*

   **Response:** We added the information how we calculated the percentage values and also added the standard deviations, which gives an estimate of the uncertainty.

   "The relative contributions of RAD and HADV to TOT vary betwen 13-29 % and 26-76 % for individual IOPs. We calculate the mean and standard deviation of the relative contributions for all IOPs and find that RAD can explain about 21±4 % and HADV about 50±17 % of the observed cooling rates on the average."

   "When calculating mean and standard deviation of the relative contribution of RAD and TURB to TOT for all IOPs – like we did in Sect. 5.2.1. – we find that each RAD and TURB explain about 22±10 % of TOT (Fig. 12). "

**References**

Babić, K., Adler, B., Kalthoff, N., Andersen, H., Dione, C., Lohou, F., Lothon, M., and Pedruzo-Bagazgoitia, X.: The observed diurnal cycle of nocturnal low-level stratus clouds over southern West Africa: a case study., Atmos. Chem. Phys., submitted, 2018.

Dione, C., Lohou, F., Lothon, M., Adler, B., Babić, K., Kalthoff, N., Pedruzo-Bagazgoitia, X., Bezombes, Y., and Gabella, O.: Intra-seasonal evolution of the most important low-troposphere dynamical structures over Southern West Africa during DACCIWA field campaign, Atmos. Chem. Phys., to be submitted, 2018.

Hannak, L., Knippertz, P., Fink, A. H., Kniffka, A., and Pante, G.: Why do global climate models

struggle to represent low-level clouds in the West African summer Monsoon?, J. Climate, 30, 1665–1687, 2017.

Knippertz, P., Fink, A. H., Deroubaix, A., Morris, E., Tocquer, F., Evans, M. J., Flamant, C., Gaetani, M., Lavaysse, C., Mari, C., et al.: A meteorological and chemical overview of the DACCIWA field campaign in West Africa in June–July 2016, Atmos. Chem. Phys., 17, 10 893–10 918, 2017.

Lothon, M., Saïd, F., Lohou, F., and Campistron, B.: Observation of the diurnal cycle in the low troposphere of West Africa, Mon. Weather Rev., 136, 3477–3500, 2008.

---

## Author Response (AR2)

**Response to the co-editor comments (ACP-2018-775)**

**"Nocturnal low-level clouds in the atmospheric boundary layer over southern West Africa: an observation-based analysis of conditions and processes"**
**by Bianca Adler et al.**

January 3, 2019

We thank the co-editor, Susan van den Heever, for her comments. We addressed all comments and provided a second revised version of the manuscript, where the new changes are highlighted in blue. Changes from the first revision are still highlighted in red.

1. *Referee 2 raised a question about IOP 10 in terms of the differences on this day when compared with others. I understand the authors arguments however, it would be useful to state something more specific to this effect including that this day was different and that these interesting differences form the basis of an upcoming manuscript.*

   **Response:** We added an explanation that IOP 10 was within the vortex period during which a drier air mass was present and mention that the differences between nights with and without clouds are the topic of another upcoming publication.

   No LLC existed at Savè during IOP 10. This IOP was within the vortex period during which a drier air mass was present in the investigation area (Kalthoff et al., 2018). A detailed analysis of the differences between nights with and without LLC is the topic of another upcoming publication.

2. *Referee 2 also asked about why the LCLs were calculated using surface conditions as opposed to using the radiosondes. This is a good question, particularly given that the fact that the radiosondes are released on a frequent basis, a point emphasized by the authors in the introduction as a strength of the field campaign, and as one of the reasons for fact for the extension of the Babic study. Can the authors please (a) include in the manuscript the reason for using the surface-based approach; and (b) comment on how the surface-based calculations of the LCL differ from those obtained using the radiosondes at the times that these are available.*

   **Response:** We are not sure that we fully understand this comment. In our study, we use the LCL to investigate the relationship between the LLC and surface processes following

the work done by Zhu et al. (2001). The LCL is only based on and defined by near-surface data (Romps, 2017). The LCL is defined as the level at which a parcel of moist air which is lifted dry-adiabatically would become saturated. On a thermodynamic diagram it is the point of intersection of the dry adiabat and constant mixing ratio line that pass through the temperature and mixing ratio of the parcel to be lifted. We calculate the LCL for a parcel lifted from the surface after the well-known approximation of Espy LCL $= 125(T - T_d)$ (e.g. given in Romps (2017)) using the continuous near-surface measurements of temperature and humidity. Information on the temperature and humidity profiles are not necessary to calculate the LCL which is why we do not use the radiosoundings for this. If we calculated the LCL from radiosoundings, we would only use the lowest measurement level of the soundings, which are basically the same as the near-surface values measured at 2 m at the tower. From the radiosoundings we could estimate the cloud base height (CBH) as the level where RH exceeds a certain threshold. This method is e.g. used in the overview paper by Kalthoff et al. (2018). As continuous information on CBH are available from the ceilometer, it is not necessary to use CBH from radiosondes in this study. We added the information that we use the approximation after Espy to calculate the LCL.

"We calculate the LCL from air temperature, $T$, and dew point temperature, $T_d$, measurements at 2 m a.g.l. with the equation after Espy LCL$= 125(T - T_d)$ (e.g. Romps, 2017), and compare it to the observed CBH. "

3. *Referees 1 and 2 both commented on the quality of the figures. I still have some concerns in this regard. More specifically:*

   a) *Figure 3 is a beautiful figure and will be most useful to readers interested in this topic. It is clear that the authors have attempted to improve this figure from the first version. However, there are still a number of issues. Firstly, the times on the abscissa have been cut off by virtue of the placement of each of the panels. Secondly, the fonts on both the left and right ordinate axes (labels and titles) are really too small to read. I tested this both with a printout and an on-screen version. Please consider ways in which to improve this. You might, for example, use just one height label on the left, applying it across panels, and one time label at the bottom (or top) and applying it across all columns.*

   **Response:** We increased the font size of the axis and colorbar bar labels as well as the indication of the IOPs. We removed the tick labels for the panels in the right column, which allows us to enlarge the individual panels. We are aware of the fact that the font size in this figure is borderline small, but we hope that in the revised version the labels are now readable.

   b) *Figure 6 has also been improved. The authors argue against making Save a solid marker, however, I think it would be clearer if it were and I dont think that it makes a difference to the cloud fields. Furthermore, the lat and lon fonts should be increased, and the times could be slightly larger too. Finally, please make a clear separation between your legend and panels. Alternatively consider using a single, much larger legend for the figure.*

**Response:** We follow the editor's suggestions and use a solid marker to indicate the location of Savè. We also increased the fonts of the labels, the times and the colorbar legend and increased the spacing between the panels and the colorbar to make a clear separation.

c) *Figure 7. None of the referees commented on figure 7, however, the label axes, as well as the legend labels are again too small on this figure. Please enlarge these fonts.*

**Response:** We enlarged the font size of the legend and axis labels.

**References**

Kalthoff, N., Lohou, F., Brooks, B., Jegede, G., Adler, B., Babić, K., Dione, C., Ajao, A., Amekudzi, L. K., Aryee, J. N. A., Ayoola, M., Bessardon, G., Danuor, S. K., Handwerker, J., Kohler, M., Lothon, M., Pedruzo-Bagazgoitia, X., Smith, V., Sunmonu, L., Wieser, A., Fink, A. H., and Knippertz, P.: An overview of the diurnal cycle of the atmospheric boundary layer during the West African monsoon season: results from the 2016 observational campaign, Atmos. Chem. Phys., 18, 2913–2928, 2018.

Romps, D. M.: Exact expression for the lifting condensation level, J. Atmos. Sci., 74, 3891–3900, 2017.

Zhu, P., Albrecht, B., and Gottschalck, J.: Formation and development of nocturnal boundary layer clouds over the southern Great Plains, J. Atmos. Sci., 58, 1409–1426, 2001.